# The genome of the crustacean *Parhyale hawaiensis,* a model for animal development, regeneration, immunity and lignocellulose digestion

Damian Kao[1†], Alvina G Lai[1†], Evangelia Stamataki[2†], Silvana Rosic[3,4], Nikolaos Konstantinides[5], Erin Jarvis[6], Alessia Di Donfrancesco[1], Natalia Pouchkina-Stancheva[1], Marie Sémon[5], Marco Grillo[5], Heather Bruce[6], Suyash Kumar[2], Igor Siwanowicz[2], Andy Le[2], Andrew Lemire[2], Michael B Eisen[7], Cassandra Extavour[8], William E Browne[9], Carsten Wolff[10], Michalis Averof[5], Nipam H Patel[6], Peter Sarkies[3,4], Anastasios Pavlopoulos[2*], Aziz Aboobaker[1*]

[1]Department of Zoology, University of Oxford, Oxford, United Kingdom; [2]Janelia Research Campus, Howard Hughes Medical Institute, Virginia, United States; [3]MRC Clinical Sciences Centre, Imperial College London, London, United Kingdom; [4]Clinical Sciences, Imperial College London, London, United Kingdom; [5]Institut de Gé nomique Fonctionnelle de Lyon, Centre National de la Recherche Scientifique (CNRS) and É cole Normale Supé rieure de Lyon, Lyon, France; [6]Department of Molecular and Cell Biology, University of California, Berkeley, United States; [7]Molecular and Cell Biology, Howard Hughes Medical Institute, University of California, Berkeley, United States; [8]Department of Organismic and Evolutionary Biology, Harvard University, Cambridge, United States; [9]Department of Invertebrate Zoology, Smithsonian National Museum of Natural History, Washington, United States; [10]Vergleichende Zoologie, Institut fur Biologie, Humboldt-Universitat zu Berlin, Berlin, Germany

**\*For correspondence:**
pavlopoulosa@janelia.hhmi.org (AP); aziz.aboobaker@zoo.ox.ac.uk (AA)

[†]These authors contributed equally to this work

**Competing interests:** The authors declare that no competing interests exist.

**Abstract** The amphipod crustacean *Parhyale hawaiensis* is a blossoming model system for studies of developmental mechanisms and more recently regeneration. We have sequenced the genome allowing annotation of all key signaling pathways, transcription factors, and non-coding RNAs that will enhance ongoing functional studies. *Parhyale* is a member of the Malacostraca clade, which includes crustacean food crop species. We analysed the immunity related genes of *Parhyale* as an important comparative system for these species, where immunity related aquaculture problems have increased as farming has intensified. We also find that *Parhyale* and other species within Multicrustacea contain the enzyme sets necessary to perform lignocellulose digestion ('wood eating'), suggesting this ability may predate the diversification of this lineage. Our data provide an essential resource for further development of *Parhyale* as an experimental model. The first malacostracan genome will underpin ongoing comparative work in food crop species and research investigating lignocellulose as an energy source.

## Introduction

Very few members of the Animal Kingdom hold the esteemed position of major model system for understanding living systems. Inventions in molecular and cellular biology increasingly facilitate the

**eLife digest** The marine crustacean known as *Parhyale hawaiensis* is related to prawns, shrimps and crabs and is found at tropical coastlines around the world. This species has recently attracted scientific interest as a possible new model to study how animal embryos develop before birth and, because *Parhyale* can rapidly regrow lost limbs, how tissues and organs regenerate. Indeed, *Parhyale* has many characteristics that make it a good model organism, being small, fast-growing and easy to keep and care for in the laboratory.

Several research tools have already been developed to make it easier to study *Parhyale*. This includes the creation of a system for using the popular gene editing technology, CRISPR, in this animal. However, one critical resource that is available for most model organisms was missing; the complete sequence of all the genetic information of this crustacean, also known as its genome, was not available.

Kao, Lai, Stamataki et al. have now compiled the *Parhyale* genome – which is slightly larger than the human genome – and studied its genetics. Analysis revealed that *Parhyale* has genes that allow it to fully digest plant material. This is unusual because most animals that do this rely upon the help of bacteria. Kao, Lai, Stamataki et al. also identified genes that provide some of the first insights into the immune system of crustaceans, which protects these creatures from diseases.

Kao, Lai, Stamataki et al. have provided a resource and findings that could help to establish *Parhyale* as a popular model organism for studying several ideas in biology, including organ regeneration and embryonic development. Understanding how *Parhyale* digests plant matter, for example, could progress the biofuel industry towards efficient production of greener energy. Insights from its immune system could also be adapted to make farmed shrimp and prawns more resistant to infections, boosting seafood production.

emergence of new experimental systems for developmental genetic studies. The morphological and ecological diversity of the phylum Arthropoda makes them an ideal group of animals for comparative studies encompassing embryology, adaptation of adult body plans and life history evolution (*Akam, 2000*; *Budd and Telford, 2009*; *Peel et al., 2005*; *Scholtz and Wolff, 2013*). While the most widely studied group are Hexapods, reflected by over a hundred sequencing projects available in the NCBI genome database, genomic data in the other three sub-phyla in Arthropoda are still relatively sparse.

Recent molecular and morphological studies have placed crustaceans along with hexapods into a pancrustacean clade (*Figure 1A*), revealing that crustaceans are paraphyletic (*Mallatt et al., 2004*; *Cook et al., 2005*; *Regier et al., 2005*; *Ertas et al., 2009*; *Richter, 2002*). Previously, the only available fully sequenced crustacean genome was that of the water flea *Daphnia* which is a member of the Branchiopoda (*Colbourne et al., 2011*). A growing number of transcriptomes for larger phylogenetic analyses have led to differing hypotheses of the relationships of the major pancrustacean groups (*Figure 1B*) (*Meusemann et al., 2010*; *Regier et al., 2010*; *Oakley et al., 2013*; *von Reumont et al., 2012*). The genome of the amphipod crustacean *Parhyale hawaiensis* addresses the paucity of high quality non-hexapod genomes among the pancrustacean group, and will help to resolve relationships within this group as more genomes and complete proteomes become available (*Rivarola-Duarte et al., 2014*; *Kenny et al., 2014*). Crucially, genome sequence data is also necessary to further advance research in *Parhyale*, currently the most tractable crustacean model system. This is particularly true for the application of powerful functional genomic approaches, such as genome editing (*Cong et al., 2013*; *Serano et al., 2015*; *Martin et al., 2015*; *Mali et al., 2013*; *Jinek et al., 2012*; *Gilles and Averof, 2014*).

*Parhyale* is a member of the diverse Malacostraca clade with thousands of extant species including economically and nutritionally important groups such as shrimps, crabs, crayfish and lobsters, as well as common garden animals like woodlice. They are found in all marine, fresh water, and higher humidity terrestrial environments. Apart from attracting research interest as an economically important food crop, this group of animals has been used to study developmental biology and the evolution of morphological diversity (for example with respect to Hox genes) (*Martin et al., 2015*;

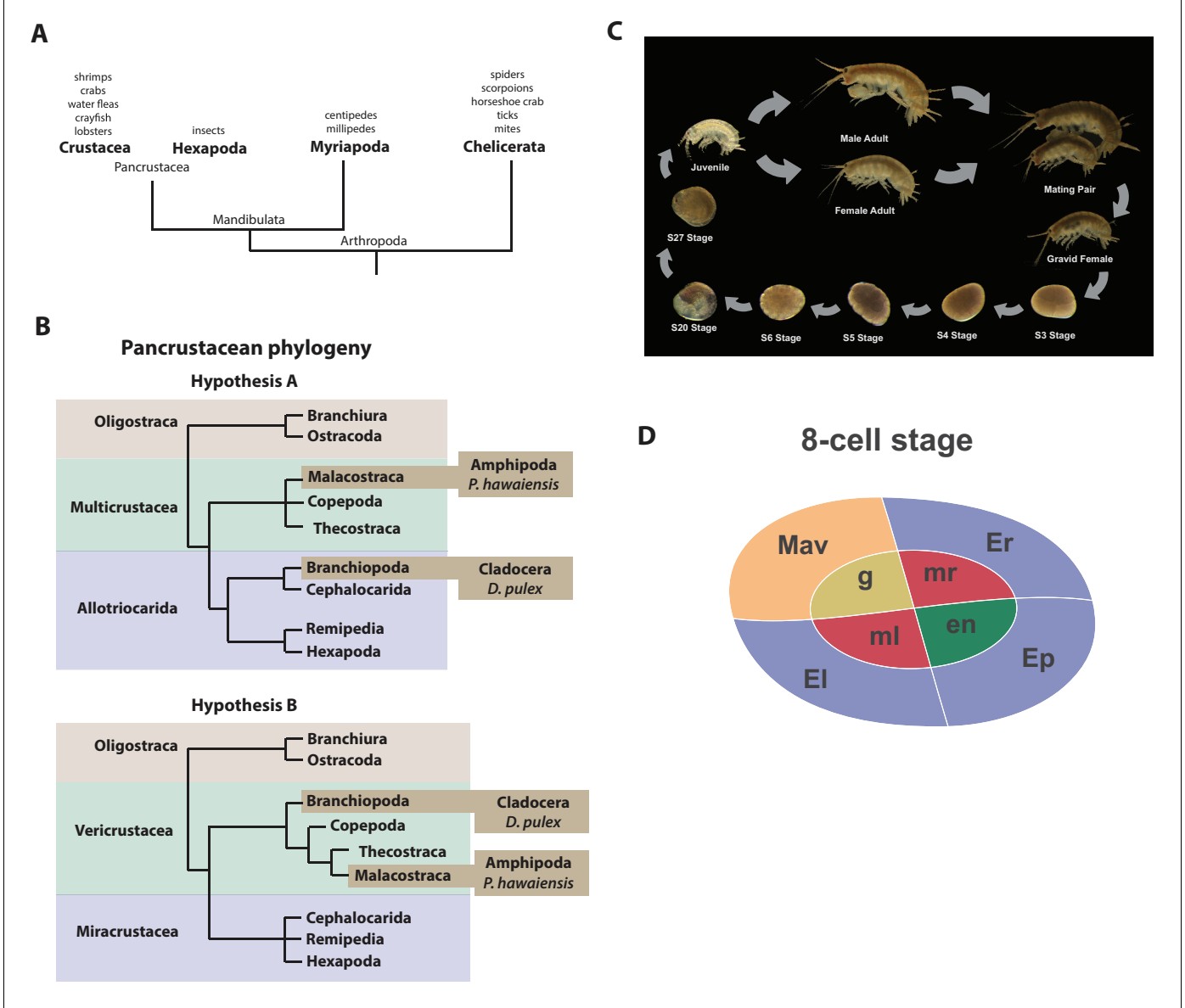

**Figure 1.** Introduction. (**A**) Phylogenetic relationship of Arthropods showing the Chelicerata as an outgroup to Mandibulata and the Pancrustacea clade which includes crustaceans and insects. Species listed for each clade have ongoing or complete genomes. Species include Crustacea: *Parhyale hawaiensis*, *D. pulex*; Hexapoda: *Drosophila melanogaster*, *Apis mellifera*, *Bombyx mori*, *Aedis aegypti*, *Tribolium castaneum*; Myriapoda: *Strigamia maritima*, *Trigoniulus corallines*; Chelicerata: *Ixodes scapularis*, *Tetranychus urticae*, *Mesobuthus martensii*, *Stegodyphus mimosarum*. (**B**) One of the unresolved issues concerns the placement of the Branchiopoda either together with the Cephalocarida, Remipedia and Hexapoda (Allotriocarida hypothesis A) or with the Copepoda, Thecostraca and Malacostraca (Vericrustacea hypothesis B). (**C**) Life cycle of *Parhyale* that takes about two months at 26C. *Parhyale* is a direct developer and a sexually dimorphic species. The fertilized egg undergoes stereotyped total cleavages and each blastomere becomes committed to a particular germ layer already at the 8-cell stage depicted in (**D**). The three macromeres Er, El, and Ep give rise to the anterior right, anterior left, and posterior ectoderm, respectively, while the fourth macromere Mav gives rise to the visceral mesoderm and anterior head somatic mesoderm. Among the 4 micromeres, the mr and ml micromeres give rise to the right and left somatic trunk mesoderm, en gives rise to the endoderm, and g gives rise to the germline.

*Averof and Patel, 1997*; *Liubicich et al., 2009*; *Pavlopoulos et al., 2009*), stem cell biology (*Konstantinides and Averof, 2014*; *Benton et al., 2014*), innate immunity processes (*Vazquez et al., 2009*; *Hauton, 2012*) and recently the cellular mechanisms of regeneration (*Konstantinides and Averof, 2014*; *Benton et al., 2014*; *Alwes et al., 2016*). In addition, members

of the Malacostraca, specifically both Amphipods and Isopods, are thought to be capable of 'wood eating' or lignocellulose digestion and to have microbiota-free digestive systems (*King et al., 2010*; *Kern et al., 2013*; *Boyle and Mitchell, 1978*; *Zimmer et al., 2002*).

The life history of *Parhyale* makes it a versatile model organism amenable to experimental manipulations (*Figure 1C*) (*Wolff and Gerberding, 2015*). Gravid females lay eggs every 2 weeks upon reaching sexual maturity and hundreds of eggs can be easily collected at all stages of embryogenesis. Embryogenesis takes about 10 days at 26°C and has been described in detail with an accurate staging system (*Browne et al., 2005*). Early embryos display an invariant cell lineage with each blastomere at the 8-cell stage contributing to a specific germ layer (*Figure 1D*) (*Browne et al., 2005*; *Gerberding et al., 2002*). Embryonic and post-embryonic stages are amenable to experimental manipulations and direct observation in vivo (*Gerberding et al., 2002*; *Extavour, 2005*; *Rehm et al., 2009a, 2009b, 2009c, 2009d*; *Price et al., 2010*; *Alwes et al., 2011*; *Hannibal et al., 2012*; *Kontarakis and Pavlopoulos, 2014*; *Nast and Extavour, 2014*; *Chaw and Patel, 2012*; *Pavlopoulos and Averof, 2005*). These can be combined with transgenic approaches (*Pavlopoulos and Averof, 2005*; *Kontarakis et al., 2011*; *Kontarakis and Pavlopoulos, 2014*; *Pavlopoulos et al., 2009*), RNA interference (RNAi) (*Liubicich et al., 2009*) and morpholino-mediated gene knockdown (*Ozhan-Kizil et al., 2009*), and transgene-based lineage tracing (*Konstantinides and Averof, 2014*). Most recently the utility of the clustered regularly interspaced short palindromic repeats (CRISPR)/CRISPR-associated (Cas) system for targeted genome editing has been elegantly demonstrated during the systematic study of *Parhyale* Hox genes (*Martin et al., 2015*; *Serano et al., 2015*). This arsenal of experimental tools (*Table 1*) has already established *Parhyale* as an attractive model system for biological research.

So far, work in *Parhyale* has been constrained by the lack of a reference genome and other standardized genome-wide resources. To address this limitation, we have sequenced, assembled and annotated the genome. At an estimated size of 3.6 Gb, this genome represents one of the largest animal genomes tackled to date. The large size has not been the only challenge of the *Parhyale* genome, that also exhibits some of the highest levels of sequence repetitiveness and polymorphism reported among published genomes. We provide information in our assembly regarding polymorphism to facilitate functional genomic approaches sensitive to levels of sequence similarity, particularly homology-dependent genome editing approaches. We analysed a number of key features of the genome as foundations for new areas of research in *Parhyale*, including innate immunity in crustaceans, lignocellulose digestion, non-coding RNA biology, and epigenetic control of the genome.

**Table 1.** Experimental resources. Available experimental resources in *Parhyale* and corresponding references.

| Experimental Resources | References |
| --- | --- |
| Embryological manipulations<br>Cell microinjection, isolation, ablation | (*Gerberding et al., 2002*; *Extavour, 2005*; *Price et al., 2010*; *Alwes et al., 2011*; *Hannibal et al., 2012*; *Rehm et al., 2009*; *Rehm et al., 2009*; *Kontarakis and Pavlopoulos, 2014*; *Nast and Extavour, 2014*) |
| Gene expression studies<br>In situ hybridization, antibody staining | (*Rehm et al., 2009*; *Rehm et al., 2009*) |
| Gene knock-down<br>RNA interference, morpholinos | (*Liubicich et al., 2009*; *Ozhan-Kizil et al., 2009*) |
| Transgenesis<br>Transposon-based, integrase-based | (*Pavlopoulos and Averof, 2005*; *Kontarakis et al., 2011*; *Kontarakis and Pavlopoulos, 2014*) |
| Gene trapping<br>Exon/enhancer trapping, iTRAC (trap conversion) | (*Kontarakis et al., 2011*) |
| Gene misexpressionHeat-inducible | (*Pavlopoulos et al., 2009*) |
| Gene knock-outCRISPR/Cas | (*Martin et al., 2015*) |
| Gene knock-in<br>CRISPR/Cas homology-dependent or homology-independent | (*Serano et al., 2015*) |
| Live imaging<br>Bright-field, confocal, light-sheet microscopy | (*Alwes et al., 2011*; *Hannibal et al., 2012*; *Chaw and Patel, 2012*; *Alwes et al., 2016*) |

Our data bring *Parhyale* to the forefront of developing model systems for a broad swathe of important bioscience research questions.

## Results and discussion

### Genome assembly, annotation, and validation

The *Parhyale* genome contains 23 pairs (2n=46) of chromosomes (*Figure 2*) and with an estimated size of 3.6 Gb, it is currently the second largest reported arthropod genome after the locust genome (*Parchem et al., 2010*; *Wang et al., 2014*). Sequencing was performed on genomic DNA isolated from a single adult male taken from a line derived from a single female and expanded after two rounds of sib-mating. We performed k-mer analyses of the trimmed reads to assess the impact of repeats and polymorphism on the assembly process. We analyzed k-mer frequencies (*Figure 3A*) and compared k-mer representation between our different sequencing libraries. We observed a 93% intersection of unique k-mers among sequencing libraries, indicating that the informational content was consistent between libraries (*Source code 1*). The k-mer analysis revealed a bimodal distribution of error-free k-mers (*Figure 3A*). The higher-frequency peak corresponded to k-mers present on both haplotypes (i.e. homozygous regions), while the lower-frequency peak had half the coverage and corresponded to k-mers present on one haplotype (i.e. heterozygous regions) (*Simpson and Durbin, 2012*). We concluded that the single sequenced adult *Parhyale* exhibits very high levels of heterozygosity, similar to the highly heterozygous oyster genome (see below).

In order to quantify global heterozygosity and repeat content of the genome we assessed the de-Bruijn graphs generated from the trimmed reads to observe the frequency of both variant and repeat branches (*Simpson, 2014*) (*Figure 3B and C*). We found that the frequency of the variant branches was 10x higher than that observed in the human genome and very similar to levels in the highly polymorphic genome of the oyster *Crassostrea gigas* (*Zhang et al., 2012*). We also observed a frequency of repeat branches approximately 4x higher than those observed in both the human and oyster genomes (*Figure 3C*), suggesting that the big size of the *Parhyale* genome can be in large part attributed to the expansion of repetitive sequences.

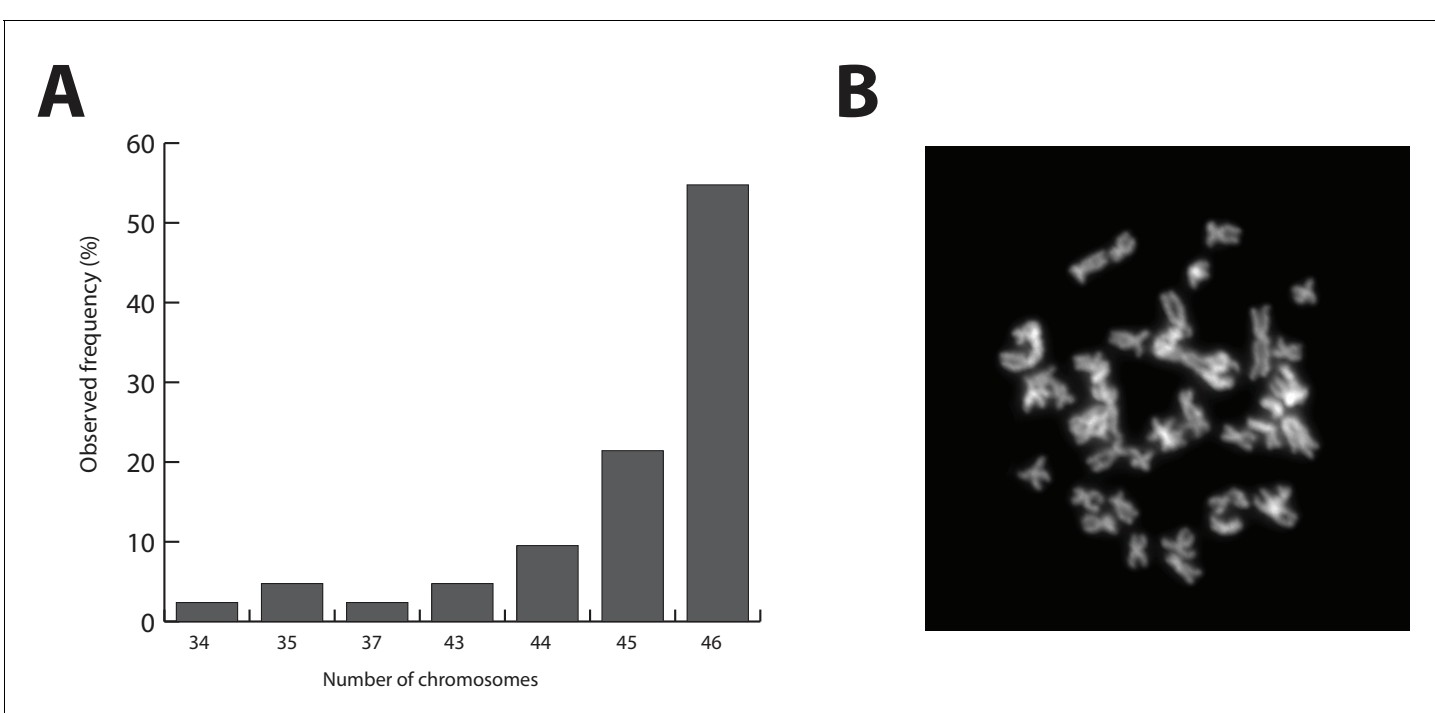

**Figure 2.** Parhyale karyotype. (**A**) Frequency of the number of chromosomes observed in 42 mitotic spreads. Forty-six chromosomes were observed in more than half of all preparations. (**B**) Representative image of Hoechst-stained chromosomes.

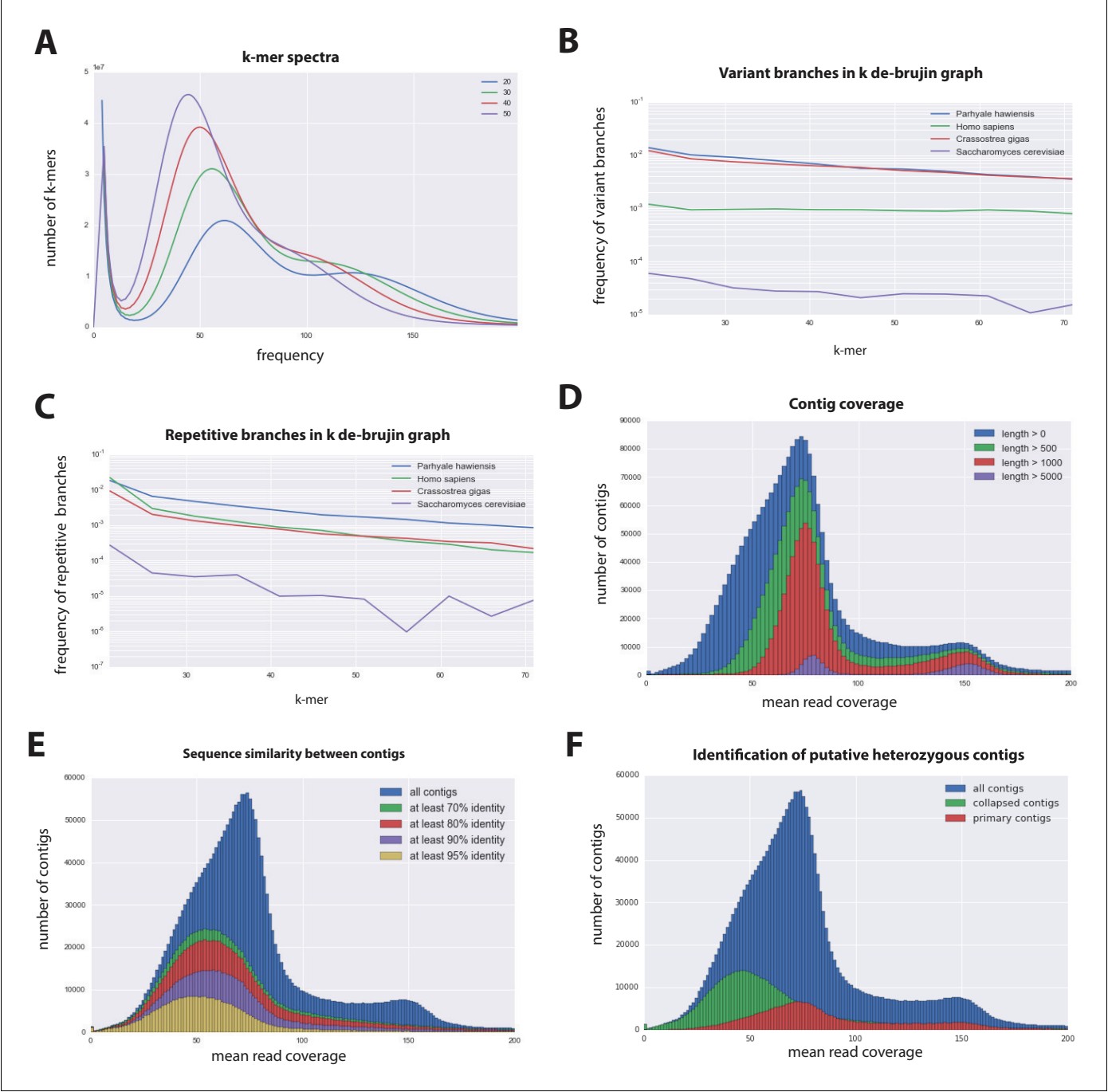

**Figure 3.** Parhyale genome assembly metrics. (**A**) K-mer frequency spectra of all reads for k-lengths ranging from 20 to 50. (**B**) K-mer branching analysis showing the frequency of k-mer branches classified as variants compared to *Homo sapiens* (human), *Crassostrea gigas* (oyster), and *Saccharomyces cerevisiae* (yeast). (**C**) K-mer branching analysis showing the frequency of k-mer branches classified as repetitive compared to *H. sapiens, C. gigas* and *S. cerevisiae*. (**D**) Histogram of read coverages of assembled contigs. (**E**) The number of contigs with an identity ranging from 70–95% to another contig in the set of assembled contigs. (**F**) Collapsed contigs (green) are contigs with at least 95% identity with a longer primary contig (red). These contigs were removed prior to scaffolding and added back as potential heterozygous contigs after scaffolding.

These metrics suggested that both contig assembly and scaffolding with mate-pair reads were likely to be challenging due to high heterozygosity and repeat content. After an initial contig assembly we remapped reads to assess coverage of each contig. We observed a major peak centered around 75x coverage and a smaller peak at 150x coverage. Contigs with lower 75x coverage represent regions of the genome that assembled into separate haplotypes and had half the frequency of mapped sequencing reads, reflecting high levels of heterozygosity. This resulted in independent assembly of haplotypes for much of the genome (*Figure 3D*).

One of the prime goals in sequencing the *Parhyale* genome was to achieve an assembly that could assist functional genetic and genomic approaches in this species. Different strategies have been employed to sequence highly heterozygous diploid genomes of non-model and wild-type samples (*Kajitani et al., 2014*). We aimed for an assembly representative of different haplotypes, allowing manipulations to be targeted to different allelic variants in the assembly. This could be particularly important for homology dependent strategies that are likely to be sensitive to polymorphism. However, the presence of alternative haplotypes could lead to poor scaffolding between contigs as many mate-pair reads may not map uniquely to one contig and distinguish between haplotypes in the assembly. To alleviate this problem we used a strategy to conservatively identify pairs of allelic contigs and proceeded to use only one in the scaffolding process. First, we estimated levels of similarity (identity and alignment length) between all assembled contigs to identify independently assembled allelic regions (*Figure 3E*). We then kept the longer contig of each pair for

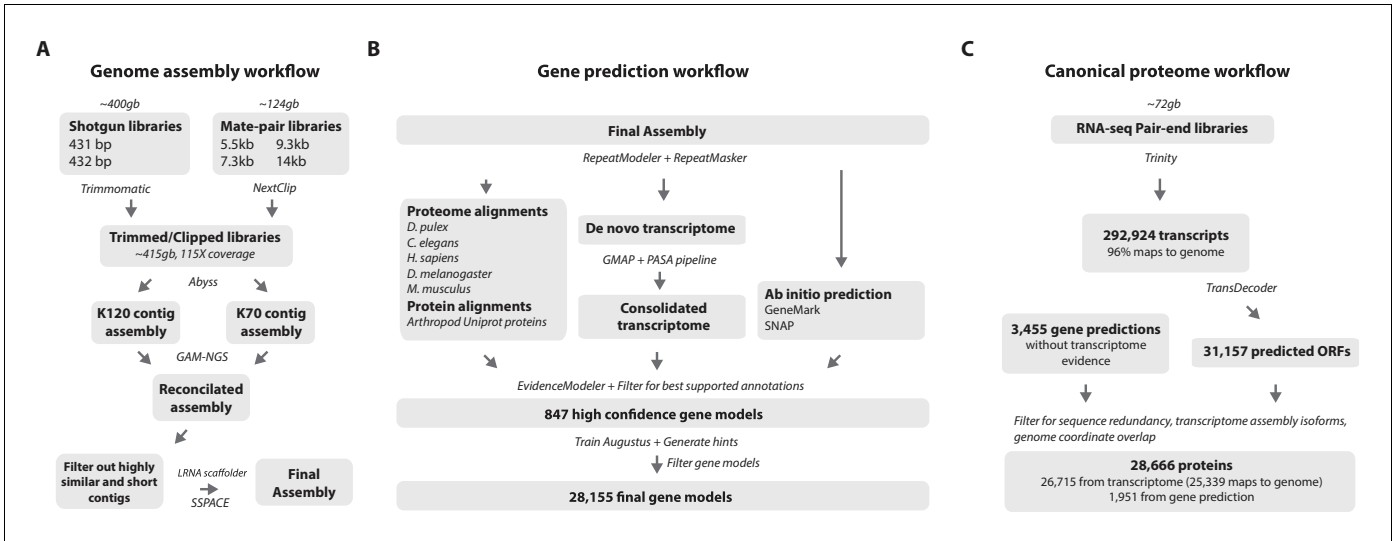

**Figure 4.** Workflows of assembly, annotation, and proteome generation. (**A**) Flowchart of the genome assembly. Two shotgun libraries and four mate-pair libraries with the indicated average sizes were prepared from a single male animal and sequenced to a predicted depth of 115x coverage after read filtering, based on a predicted size of 3.6 Gbp. Contigs were assembled at two different k-lengths with Abyss and the two assemblies were merged with GAM-NGS. Filtered contigs were scaffolded with SSPACE. (**B**) The final scaffolded assembly was annotated with a combination of Evidence Modeler to generate 847 high quality gene models and Augustus for the final set of 28,155 predictions. These protein-coding gene models were generated based on a *Parhyale* transcriptome consolidated from multiple developmental stages and conditions, their homology to the species indicated, and *ab initio* predictions with GeneMark and SNAP. (**C**) The *Parhyale* proteome contains 28,666 entries based on the consolidated transcriptome and gene predictions. The transcriptome contains 292,924 coding and non-coding RNAs, 96% of which could be mapped to the assembled genome.

The following source data and figure supplement are available for figure 4:

**Source data 1.** Catalog of repeat elements in *Parhyale* genome assembly.
**Source data 2.** Software and Data.
**Figure supplement 1.** CEGMA assessment of Parhyale transcriptome and genome.

scaffolding using our mate-pair libraries (*Figure 3F*), after which we added back the shorter allelic contigs to produce the final genome assembly (*Figure 4A*).

RepeatModeler and RepeatMasker were used on the final assembly to find repetitive regions, which were subsequently classified into families of transposable elements or short tandem repeats (*Source code 2*). We found 1473 different repeat element sequences representing 57% of the assembly (*Figure 4—source data 1*). The *Parhyale* assembly comprises of 133,035 scaffolds (90% of assembly), 259,343 unplaced contigs (4% of assembly), and 584,392 shorter, potentially allelic contigs (6% of assembly), with a total length of 4.02 Gb (*Table 2*). The N50 length of the scaffolds is 81,190 bp. The final genome assembly was annotated with Augustus trained with high confidence gene models derived from assembled transcriptomes, gene homology, and *ab initio* predictions. This resulted in 28,155 final gene models (*Figure 4B*; *Source code 3*) across 14,805 genic scaffolds and 357 unplaced contigs with an N50 of 161,819, bp and an N90 of 52,952 bp.

*Parhyale* has a mean coding gene size (introns and ORFs) of 20 kb (median of 7.2 kb), which is longer than *D. pulex* (mean: 2 kb, median: 1.2 kb), while shorter than genes in *Homo sapiens* (mean: 52.9 kb, median: 18.5 kb). This difference in gene length was consistent across reciprocal blast pairs where ratios of gene lengths revealed *Parhyale* genes were longer than *Caenorhabditis elegans*, *D. pulex*, and *Drosophila melanogaster* and similar to *H. sapiens*. (*Figure 5A*). The mean intron size in *Parhyale* is 5.4 kb, similar to intron size in *H. sapiens* (5.9 kb) but dramatically longer than introns in *D. pulex* (0.3 kb), *D. melanogaster* (0.3 kb) and *C. elegans* (1 kb) (*Figure 5B*).

For downstream analyses of *Parhyale* protein coding content, a final proteome consisting of 28,666 proteins was generated by combining candidate coding sequences identified with TransDecoder (*Haas et al., 2013*) from mixed stage transcriptomes. Almost certainly the high number of predicted gene models and proteins is an overestimation due to fragmented genes, very different isoforms or unresolved alleles, that will be consolidated as annotation of the *Parhyale* genome improves. We also included additional high confidence gene predictions that were not found in the transcriptome (*Figure 4C*). The canonical proteome dataset was annotated with both Pfam, KEGG, and BLAST against Uniprot. Assembly quality was further evaluated by alignment to core eukaryotic genes defined by the Core Eukaryotic Genes Mapping Approach (CEGMA) database (*Parra et al., 2007*). We identified 244/248 CEGMA orthology groups from the assembled genome alone and 247/248 with a combination of genome and mapped transcriptome data (*Figure 4—figure supplement 1*). Additionally, 96% of over 280,000 identified transcripts, most of which are fragments that do not contain a large ORF, also mapped to the assembled genome. Together these data suggest that our assembly is close to complete with respect to protein coding genes and transcribed regions that are captured by deep RNA sequencing.

## High levels of heterozygosity and polymorphism in the *Parhyale* genome

To estimate the level of heterozygosity in genes we first identified transcribed regions of the genome by mapping back transcripts to the assembly. Where these regions appeared in a single contig in the assembly, heterozygosity was calculated using information from mapped reads. Where these regions appeared in more than one contig, because haplotypes had assembled independently, heterozygosity was calculated using an alignment of the genomic sequences corresponding to mapped transcripts and information from mapped reads. This allowed us to calculate heterozygosity for each gene within the sequenced individual (*Source code 4*). We then calculated the genomic coverage of all transcribed regions in the genome and found, as expected, they fell broadly into two

**Table 2.** Assembly statistics. Length metrics of assembled scaffolds and contigs.

|  | # sequences | N90 | N50 | N10 | Sum length | Max length | # Ns |
|---|---|---|---|---|---|---|---|
| scaffolds | 133,035 | 14,799 | 81,190 | 289,705 | 3.63 GB | 1,285,385 | 1.10 GB |
| unplaced contigs | 259,343 | 304 | 627 | 1779 | 146 MB | 40,222 | 23,431 |
| hetero. contigs | 584,392 | 265 | 402 | 1038 | 240 MB | 24,461 | 627 |
| genic scaffolds | 15,160 | 52952 | 161,819 | 433836 | 1.49 GB | 1,285,385 | 323 MB |

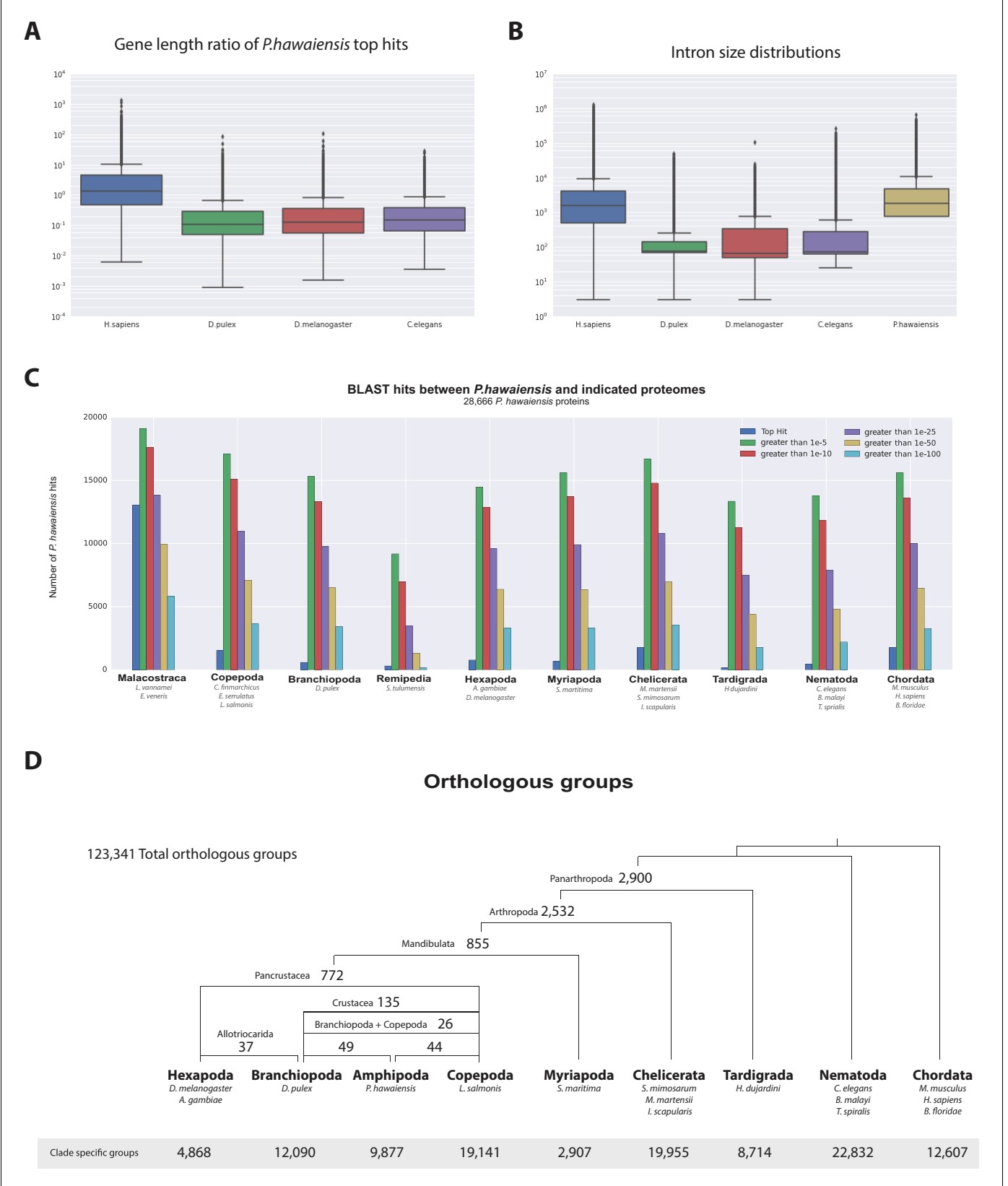

**Figure 5.** Parhyale genome comparisons. (**A**) Box plots comparing gene sizes between *Parhyale* and humans (*H. sapiens*), water fleas (*D. pulex*), flies (*D. melanogaster*) and nematodes (*C. elegans*). Ratios were calculated by dividing the size of the top blast hit in each species with the corresponding
*Figure 5 continued on next page*

*Figure 5 continued*

*Parhyale* gene size. (B) Box plots showing the distribution of intron sizes in the same species used in A. (C) Comparison between *Parhyale* and representative proteomes from the indicated animal taxa. Colored bars indicate the number of blast hits recovered across various thresholds of E-values. The top hit value represents the number of proteins with a top hit corresponding to the respective species. (D) Cladogram showing the number of shared orthologous protein groups at various taxonomic levels, as well as the number of clade-specific groups. A total of 123,341 orthogroups were identified with Orthofinder across the 16 genomes used in this analysis. Within Pancrustacea, 37 orthogroups were shared between Branchiopoda and Hexapoda (supporting the Allotriocarida hypothesis) and 49 orthogroups were shared between Branchiopoda and Amphipoda (supporting the Vericrustacea hypothesis).

The following source data and figure supplement are available for figure 5:

**Source data 1.** List of proteins currently unique to *Parhyale*.

**Source data 2.** List of genes likely to be specific to the Malacostraca

**Source data 3.** Orthofinder analysis.

**Figure supplement 1.** Expanded gene families in *Parhyale*.

categories with higher and lower read coverage (*Figure 6A*; *Source code 4*). Genes that fell within the higher read coverage group had a lower mean heterozygosity (1.09% of bases displaying polymorphism), which is expected as more reads were successfully mapped. Genes that fell within the lower read coverage group had a higher heterozygosity (2.68%), as reads mapped independently to each haplotype (*Figure 6B*) (*Simpson, 2014*). Thus, we conclude that heterozygosity that influences read mapping and assembly of transcribed regions, and not just non-coding parts of the assembly.

The assembled *Parhyale* transcriptome was derived from various laboratory populations, hence we expected to see additional polymorphism beyond that detected in the two haplotypes of the individual male we sequenced. Analysing all genes using the transcriptome we found additional variations in transcribed regions not found in the genome of the sequenced individual. In addition to polymorphisms that agreed with heterozygosity in the genome sequence we observed that the rate of additional variations is not substantially different between genes from the higher (0.88%) versus lower coverage group genes (0.73%; *Figure 6C*). This analysis suggests that within captive laboratory populations of *Parhyale* there is considerable additional polymorphism distributed across genes, irrespective of whether or not they have relatively low or high heterozygosity in the individual male we sequenced. In addition the single male we have sequenced provides an accurate reflection of polymorphism of the wider laboratory population and the established Chicago-F strain does not by chance contain unusually divergent haplotypes. We also performed an assessment of polymorphism on previously cloned *Parhyale* developmental genes, and found some examples of startling levels of variation. (*Figure 6—figure supplement 1* and source data 1). For example, we found that the cDNAs of the germ line determinants, *nanos* (78 SNPS, 34 non-synonymous substitutions and one 6 bp indel) and *vasa* (37 SNPs, 7 non-synonymous substitutions and a one 6 bp indel) can have more variability within laboratory *Parhyale* populations than might be observed for orthologs between closely related species (*Figure 6—source data 1*).

To further evaluate the extent of polymorphism across the genome, we mapped the genomic reads to a set of previously Sanger-sequenced BAC clones of the *Parhyale* Hox cluster from the same Chicago-F line from which we sequenced the genome of an adult male. (*Serano et al., 2015*). We detected SNPs at a rate of 1.3 to 2.5% among the BACs (*Table 3*) and also additional sequence differences between the BACs and genomic reads, confirming that additional polymorhism exists in the Chicago-F line beyond that detected between in the haplotypes of the individual male we sequenced.

Overlapping regions of the contiguous BACs gave us the opportunity to directly compare Chicago-F haplotypes and accurately observe polynucleotide polymorphisms, that are difficult to detect with short reads that do not map when polymorphisms are large, but are resolved by longer Sanger reads. (*Figure 7A*). Since the BAC clones were generated from a pool of Chicago-F animals, we expected each sequenced BAC to be representative of one haplotype. Overlapping regions

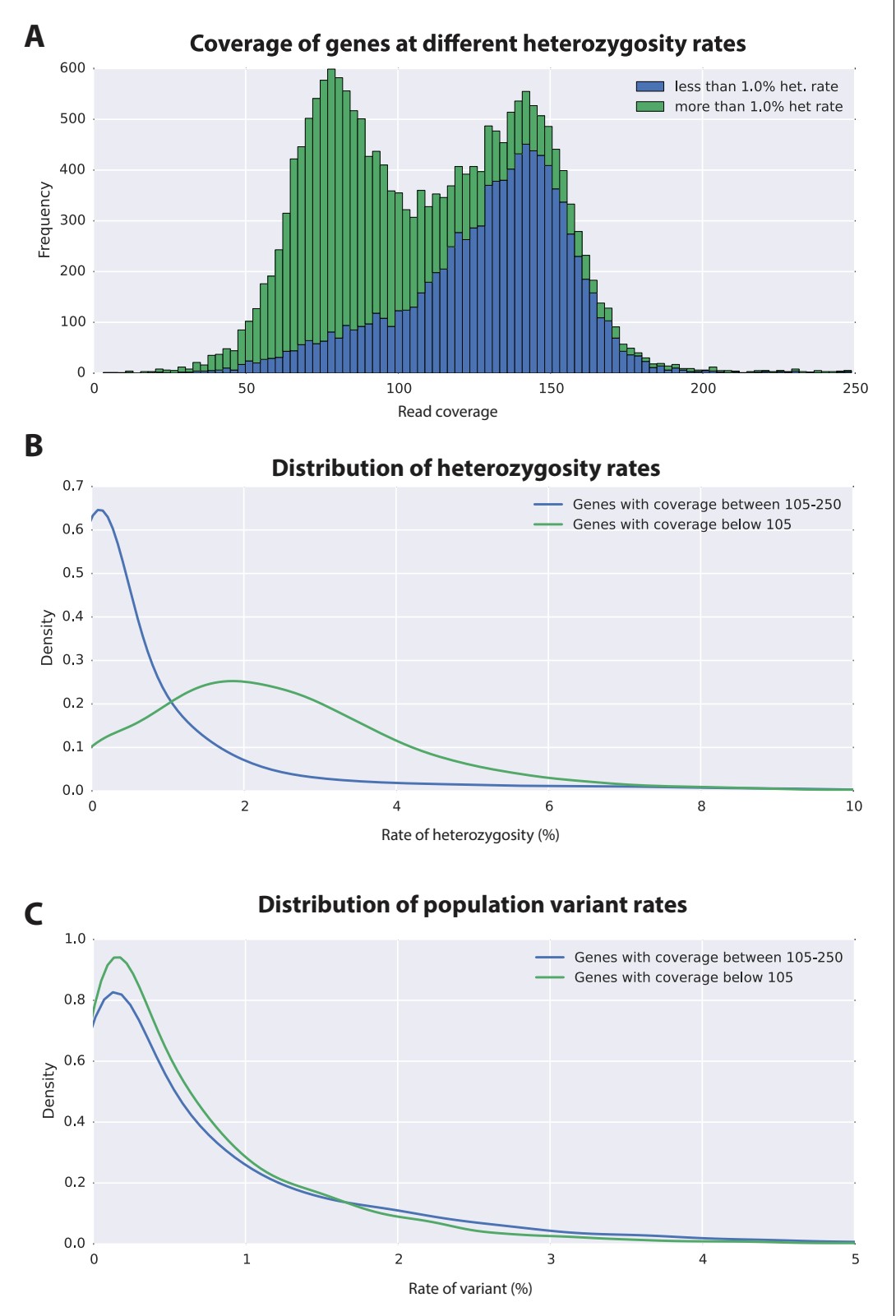

**Figure 6.** Variation analyses of predicted genes. (**A**) A read coverage histogram of predicted genes. Reads were first mapped to the genome, then coverage was calculated for transcribed regions of each defined locus. (**B**) A coverage distribution plot showing that genes in the lower coverage region (<105x coverage, peak at 75x ) have a higher level of heterozygosity than genes in the higher coverage region (>105 coverage and <250, peak at
*Figure 6 continued on next page*

*Figure 6 continued*
approximately 150x coverage). (**C**) Distribution plot indicating that mean level of population variance is similar for genes in the higher and lower coverage regions.
The following source data and figure supplement are available for figure 6:
**Source data 1.** Polymorphism in *Parhyale* devlopmental genes.
**Figure supplement 1.** Confirmation of polymorphisms in the wider laboratory population of *Parhyale*.

between BAC clones could potentially represent one or two haplotypes. We found that the genomic reads supported the SNPs observed between the overlapping BAC regions. We found relatively few base positions with evidence supporting the existence of a third allele. This analysis revealed many insertion/deletion (indels) with some cases of indels larger than 100 base pairs (*Figure 7B*). The finding that polynucleotide polymorphisms are prevalent between the haplotypes of the Chicago-F is another reason, in addition to regions of high SNP heterozygosity in the genome sequence, for the extensive independent assembly of haplotypes. Taken togther these data mean that special attention will have to be given to those functional genomic approaches that are dependent on homology, such as CRISPR/Cas9 based knock in strategies.

## A comparative genomic analysis of the *Parhyale* genome

Assessment of conservation of the proteome using BLAST against a selection of metazoan proteomes was congruent with broad phylogenetic expectations. These analyses included crustacean proteomes likely to be incomplete as they come from limited transcriptome datasets, but nonetheless highlighted genes likely to be specific to the Malacostraca (*Figure 5C*, *Figure 5—source data 2*). To better understand global gene content evolution we generated clusters of orthologous and paralogous gene families comparing the *Parhyale* proteome with other complete proteomes across the Metazoa using Orthofinder (*Emms and Kelly, 2015*) (*Figure 5D*; *Figure 5—source data 3*). Amongst proteins conserved in protostomes and deuterostomes we saw no evidence for

**Table 3.** BAC variant statistics. Level of heterozygosity of each BAC sequence determined by mapping genomic reads to each BAC individually. Population variance rate represents additional alleles found (i.e. more than 2 alleles) from genomic reads.

| BAC ID | Length | Heterozygosity | Pop.Variance |
| --- | --- | --- | --- |
| PA81-D11 | 140,264 | 1.654 | 0.568 |
| PA40-O15 | 129,957 | 2.446 | 0.647 |
| PA76-H18 | 141,844 | 1.824 | 0.199 |
| PA120-H17 | 126,766 | 2.673 | 1.120 |
| PA222-D11 | 128,542 | 1.344 | 1.404 |
| PA31-H15 | 140,143 | 2.793 | 0.051 |
| PA284-I07 | 141,390 | 2.046 | 0.450 |
| PA221-A05 | 148,703 | 1.862 | 1.427 |
| PA93-L04 | 139,955 | 2.177 | 0.742 |
| PA272-M04 | 134,744 | 1.925 | 0.982 |
| PA179-K23 | 137,239 | 2.671 | 0.990 |
| PA92-D22 | 126,848 | 2.650 | 0.802 |
| PA268-E13 | 135,334 | 1.678 | 1.322 |
| PA264-B19 | 108,571 | 1.575 | 0.157 |
| PA24-C06 | 141,446 | 1.946 | 1.488 |

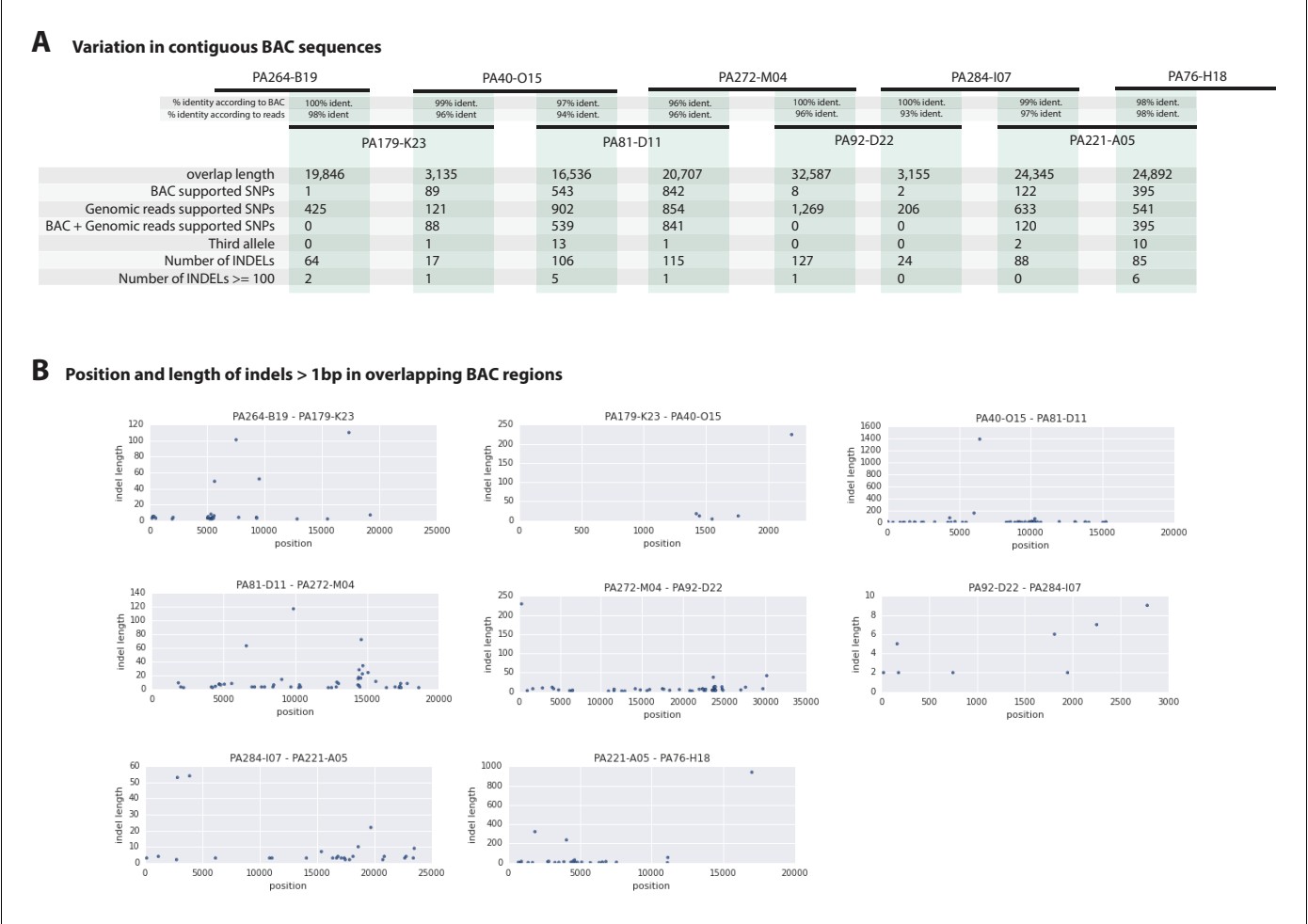

**Figure 7.** Variation observed in contiguous BAC sequences. (**A**) Schematic diagram of the contiguous BAC clones tiling across the HOX cluster and their% sequence identities. 'Overlap length' refers to the lengths (bp) of the overlapping regions between two BAC clones. 'BAC supported single nucleotide polymorphisms (SNPs)' refer to the number of SNPs found in the overlapping regions by pairwise alignment.'Genomic reads supported SNPs' refer to the number of SNPs identified in the overlapping regions by mapping all reads to the BAC clones and performing variant calling with GATK. 'BAC + Genomic reads supported SNPs' refer to the number of SNPs identified from the overlapping regions by pairwise alignment that are supported by reads. 'Third allele' refers to presence of an additional polymorphism not detected by genomic reads. 'Number of INDELs' refer to the number of all insertion or deletions found in the contiguous region. 'Number of INDELs >100' are insertion or deletions greater than or equal to 100. (**B**) Position versus indel lengths across each overlapping BAC region.

widespread gene duplication in the lineage leading to *Parhyale*. We identified orthologous and paralogous protein groups across 16 species with 2900 and 2532 orthologous groups containing proteins found only in Panarthropoda and Arthropoda respectively. We identified 855 orthologous groups that were shared exclusively by Mandibulata, 772 shared by Pancrustacea and 135 shared by Crustacea. There were 9877 *Parhyale* proteins that could not be assigned to an orthologous group, potentially representing rapidly evolving or lineage specific proteins (*Figure 5—source data 1*). Amongst these proteins we found 609 proteins (2.1% of the proteome) that had paralogs within *Parhyale*, suggesting that younger and/or more divergent *Parhyale* genes have undergone some considerable level of gene duplication events.

Our analysis of shared orthologous groups was equivocal with regard to alternative hypotheses on the relationships among pancrustacean subgroups: 44 groups of orthologous proteins are shared among the multicrustacea clade (uniting the Malacostraca, Copepoda and Thecostraca), 37 groups are shared among the Allocarida (Branchiopoda and Hexapoda) and 49 groups are shared among the Vericrustacea (Branchiopoda and Multicrustacea.

To further analyse the evolution of the *Parhyale* proteome we examined protein families that appeared to be expanded (z-score >2), compared to other taxa (*Figure 5—figure supplement 1*, *Source code 5*). We conservatively identified 29 gene families that are expanded in *Parhyale*. Gene family expansions include the Sidestep (55 genes) and Lachesin (42) immunoglobulin superfamily proteins as well as nephrins (33 genes) and neurotrimins (44 genes), which are thought to be involved in immunity, neural cell adhesion, permeability barriers and axon guidance (*Strigini et al., 2006*; *Garver et al., 2008*; *Siebert et al., 2009*). Other *Parhyale* gene expansions include *APN* (aminopeptidase N) (38 genes) and cathepsin-like genes (30 genes), involved in proteolytic digestion (*Deraison et al., 2004*).

## Major signaling pathways and transcription factors in *Parhyale*

Components of all common metazoan cell-signalling pathways are largely conserved in *Parhyale*. At least 13 *Wnt* subfamilies were present in the cnidarian-bilaterian ancestor. *Wnt3* has been lost in protostomes that retain 12 *Wnt* genes (*Prud'homme et al., 2002*; *Cho et al., 2010-07*; *Janssen et al., 2010*). Some sampled ecdysozoans have undergone significant *Wnt* gene loss, for example *C. elegans* has only 5 *Wnt* genes (*Hilliard and Bargmann, 2006*). At most 9 *Wnt* genes are present in any individual hexapod species (*Bolognesi et al., 2008*), with *wnt2* and *wnt4* potentially lost before the hexapod radiation (*Hogvall et al., 2014*). The *Parhyale* genome encodes 6 of the 13 *Wnt* subfamily genes; *wnt1, wnt4, wnt5, wnt10, wnt11* and *wnt16* (*Figure 8*). *Wnt* genes are known to have been ancestrally clustered (*Holstein, 2012*). We observed that *wnt1* and *wnt10* are linked in a single scaffold (phaw_30.0003199); given the loss of *wnt6* and *wnt9*, this may be the remnant of the ancient *wnt9-1-6-10* cluster conserved in some protostomes.

We could identify 2 Fibroblast Growth Factor (*FGF*) genes and only a single FGF receptor (*FGFR*) in the *Parhyale* genome, suggesting one *FGFR* has been lost in the malacostracan lineage (*Figure 8——figure supplement 1*). Within the Transforming Growth Factor beta (*TGF-β*) signaling pathway we found 2 genes from the activin subfamily (an activin receptor and a myostatin), 7 genes from the Bone Morphogen Protein (*BMP*) subfamily and 2 genes from the inhibin subfamily. Of the *BMP* genes, *Parhyale* has a single decapentaplegic homologue (*Figure 8—source data 2*). Other components of the TGF-β pathway were identified such as the neuroblastoma suppressor of tumorigenicity (NBL1/DAN), present in *Aedes aegypti* and *Tribolium castaneum* but absent in *D. melanogaster* and *D. pulex*, and TGFB-induced factor homeobox 1 (*TGIF1*) which is a Smad2-binding protein within the pathway present in arthropods but absent in nematodes (*C. elegans* and *Brugia malayi*; *Figure 8—source data 2*). We identified homologues of *PITX2*, a downstream target of the TGF-β pathway involved in endoderm and mesoderm formation present in vertebrates and crustaceans (*Parhyale* and *D. pulex*) but not in insects and nematodes (*Ryan et al., 1998*). With the exception of *SMAD7* and *SMAD8/9*, all other *SMADs* (*SMAD1, SMAD2/3, SMAD4, SMAD6*) are found in arthropods sampled, including *Parhyale*. Components of other pathways interacting with TGF-β signaling like the *JNK, Par6, ROCK1/RhoA, p38* and *Akt* pathways were also recovered and annotated in the *Parhyale* genome (*Figure 8—source data 2*). We identified major Notch signaling components including Notch, Delta, Deltex, Fringe and modulators of the Notch pathway such as *Dvl* and *Numb*. Members of the gamma-secretase complex (Nicastrin, Presenilin, and *APH1*) were also present as well as to other co-repressors of the Notch pathway such as Groucho and *CtBP* (*Nagel et al., 2005*).

A genome wide survey to annotate all potential transcription factors (TFs) discovered a total of 1143 proteins with DNA binding domains that belonged to all the major families previously identified. Importantly, we observed a large expansion of TFs containing the zinc-finger (ZF)-C2H2 domain, that was previously observed in a trancriptomic study of *Parhyale* (*Zeng et al., 2011*). *Parhyale* has 699 ZF-C2H2-containing genes (Chung et al., 2002–12], which is comparable to the number found in *H. sapiens* (*Najafabadi et al., 2015*), but significantly expanded compared to other arthropod species like *D. melanogaster* encoding 326 members (*Figure 8—source data 1*).

The *Parhyale* genome contains 126 homeobox-containing genes (*Figure 9*; *Figure 8—source data 3*), which is higher than the numbers reported for other arthropods (104 genes in *D. melanogaster*, 93 genes in the honey bee *Apis melllifera*, and 113 in the centipede *Strigamia maritima*) (*Chipman et al., 2014*). We identified a *Parhyale* specific expansion in the Ceramide Synthase (*CERS*) homeobox proteins, which include members with divergent homeodomains (*Pewzner-Jung et al., 2006*). *H. sapiens* have six *CERS* genes, but only five with homeodomains

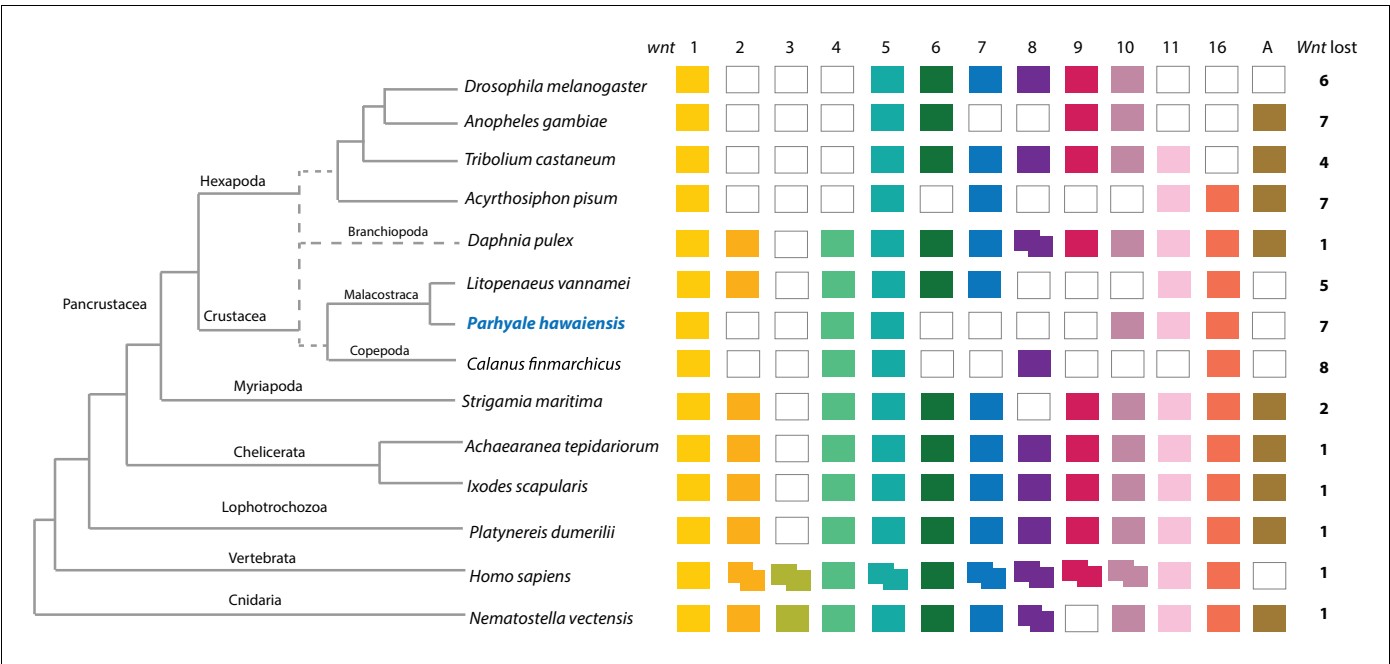

**Figure 8.** Comparison of Wnt family members across Metazoa. Comparison of Wnt genes across Metazoa. Tree on the left illustrates the phylogenetic relationships of species used. Dotted lines in the phylogenetic tree illustrate the alternative hypothesis of Branchiopoda + Hexapoda versus Branchiopoda + Multicrustacea. Colour boxes indicate the presence of certain Wnt subfamily members (wnt1 to wnt11, wnt16 and wntA) in each species. Empty boxes indicate the loss of particular Wnt genes. Two overlapping colour boxes represent duplicated Wnt genes.

The following source data and figure supplements are available for figure 8:

**Source data 1.** List of *Parhyale* transcription factors by family.

**Source data 2.** Wnt, TGFβ and FGF signaling pathways .

**Source data 3.** Homeobox transcription factors.

**Figure supplement 1.** Phylogenetic tree of FGF and FGR molecules

**Figure supplement 2.** Phylogenetic tree of CERS homeobox family genes.

(*Holland et al., 2007*). We observed an expansion to 12 *CERS* genes in *Parhyale*, compared to 1–4 genes found in other arthropods (*Zhong and Holland, 2011*) (*Figure 8—figure supplement 2*). In phylogenetic analyses all 12 *CERS* genes in *Parhyale* clustered together with a *CERS* from another amphipod *Echinogammarus veneris*, suggesting that this is recent expansion in the amphipod lineage.

*Parhyale* contains a complement of 9 canonical Hox genes that exhibit both spatial and temporal colinearity in their expression along the anterior-posterior body axis (*Serano et al., 2015*). Chromosome walking experiments had shown that the Hox genes *labial* (*lab*) and *proboscipedia* (*pb*) are linked and that *Deformed* (*Dfd*), *Sex combs reduced* (*Scr*), *Antennapedia* (*Antp*) and *Ultrabithorax* (*Ubx*) are also contiguous in a cluster (*Serano et al., 2015*). Previous experiments in *D. melanogaster* had shown that the proximity of nascent transcripts in RNA fluorescent in situ hybridizations (FISH) coincide with the position of the corresponding genes in the genomic DNA (*Kosman et al., 2004*; *Ronshaugen and Levine, 2004*). Thus, we obtained additional information on Hox gene linkage by examining nascent Hox transcripts in cells where Hox genes are co-expressed. We first validated this methodology in *Parhyale* embryos by confirming with FISH, the known linkage of *Dfd* with *Scr* in the first maxillary segment where they are co-expressed (*Figure 10A–A''*). As a negative control, we detected no linkage between *engrailed1* (*en1*) and *Ubx* or *abd-A* transcripts (*Figure 10B - B'',C -*

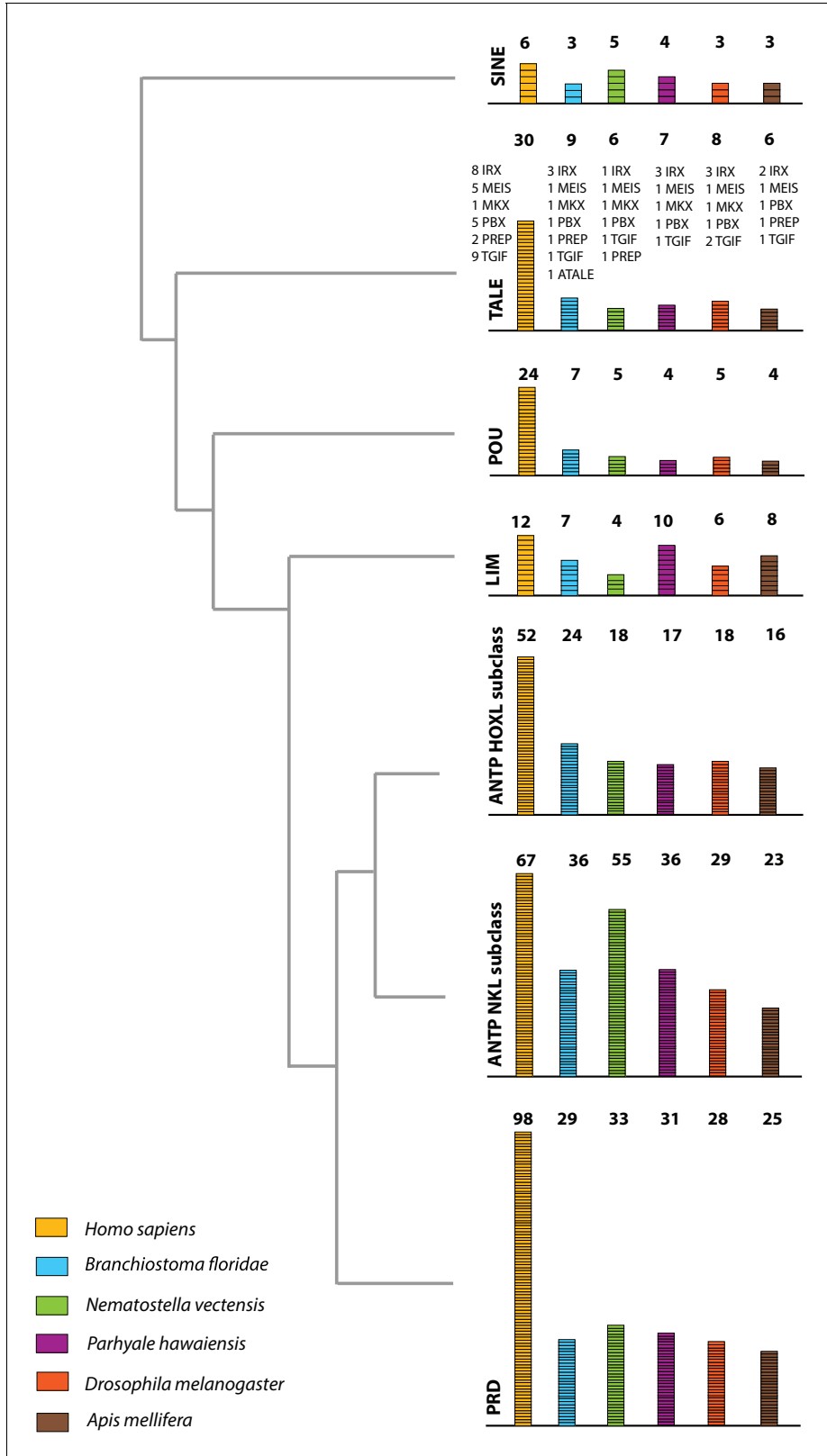

**Figure 9.** Homeodomain protein famil y t ree. The overview of homeodomain radiation and phylogenetic relationships among homeodomain proteins from Arthropoda (*P. hawaiensis, D. melanogaster and A. mellifera*), Chordata (*H. sapiens and B. floridae*), and Cnidaria (*N. vectensis*). Six major homeodomain classes are illustrated (SINE, TALE, POU, LIM, ANTP and PRD) with histograms indicating the number of genes in each species belonging to a given class.

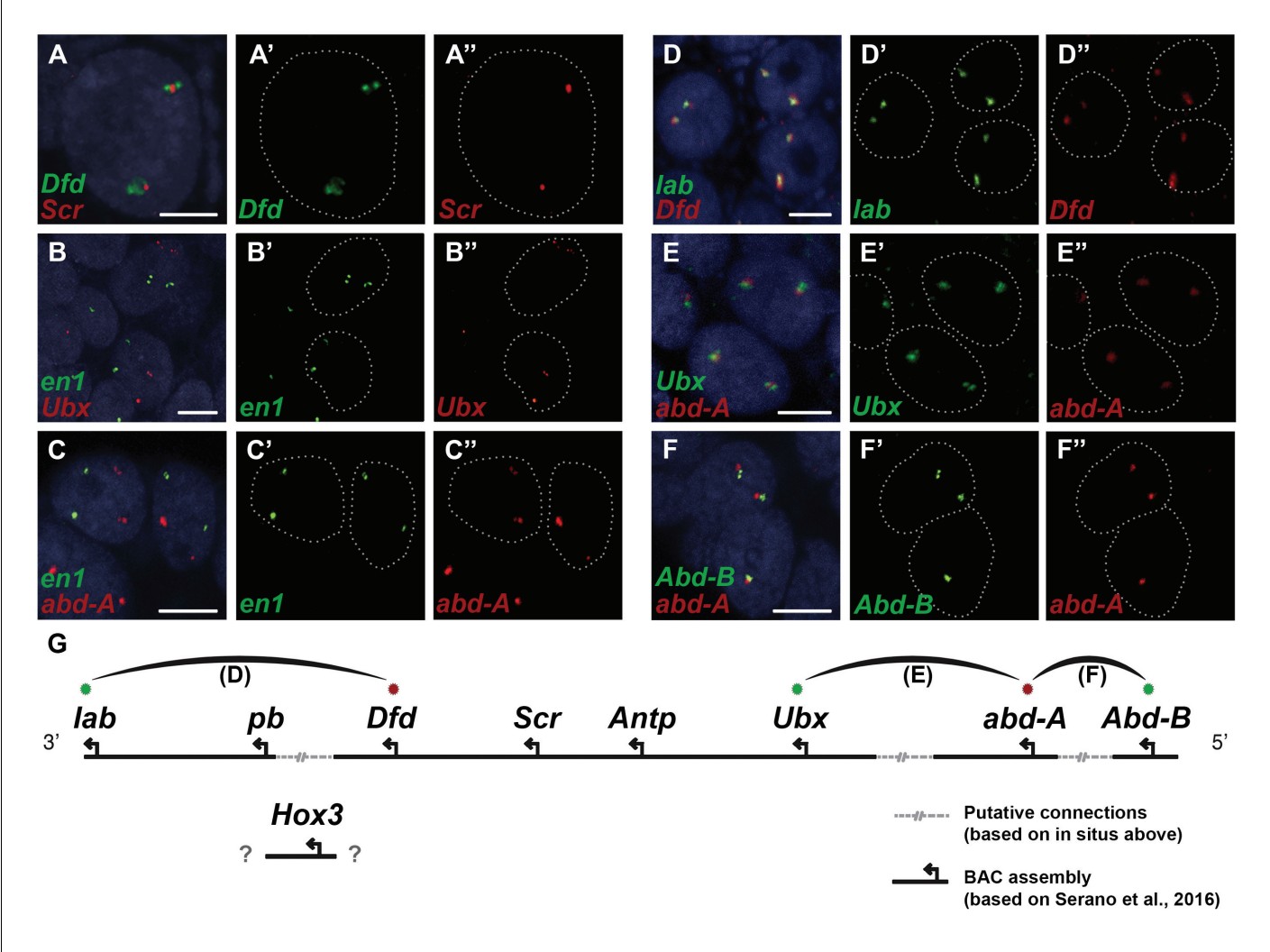

**Figure 10.** Evidence for an intact Hox cluster in Parhyale. (A–F'') Double fluorescent in situ hybridizations (FISH) for nascent transcripts of genes. (A–A'') Deformed (Dfd) and Sex combs reduced (Scr), (B-B'') engrailed 1 (en1) and Ultrabithorax (Ubx), (C–C'') en1 and abdominal-A (abd-A), (D–D'') labial (lab) and Dfd, (E–E'') Ubx and abd-A, and (F–F'') Abdominal-B (Abd-B) and abd-A. Cell nuclei are stained with DAPI (blue) in panels A–F and outlined with white dotted lines in panels A'–F' and A''. Co-localization of nascent transcript dots in A, D, E and F suggest the proximity of the corresponding Hox genes in the genomic DNA. As negative controls, the en1 nascent transcripts in B and C do not co-localize with those of Hox genes Ubx or abd-A. (G) Schematic representation of the predicted configuration of the Hox cluster in Parhyale. Previously identified genomic linkages are indicated with solid black lines, whereas linkages established by FISH are shown with dotted gray lines. The arcs connecting the green and red dots represent the linkages identified in D, E and F, respectively. The position of the Hox3 gene is still uncertain. Scale bars are 5 µm.

*C''*). We then demonstrated the tightly coupled transcripts of *lab* with *Dfd* (co-expressed in the second antennal segment, *Figure 10D - D''*), *Ubx* and *abd-A* (co-expressed in the posterior thoracic segments, *Figure 10E - E''*), and *abd-A* with *Abd-B* (co-expressed in the anterior abdominal segments, (*Figure 10F - F''*). Collectively, all evidence supports the linkage of all analysed Hox genes into a single cluster as shown in (*Figure 10G - G''*). The relative orientation and distance between certain Hox genes still needs to be worked out. So far, we have not been able to confirm that *Hox3* is also part of the cluster due to the difficulty in visualizing nascent transcripts for *Hox3* together with *pb* or *Dfd*. Despite these caveats, *Parhyale* provides an excellent arthropod model system to understand these still enigmatic phenomena of Hox gene clustering and spatio-temporal colinearity, and compare the underlying mechanisms to other well-studied vertebrate and invertebrate models (*Kmita and Duboule, 2003*).

The ParaHox and *NK* gene clusters encode other *ANTP* class homeobox genes closely related to Hox genes (*Brooke et al., 1998*). In *Parhyale*, we found 2 caudal (*Cdx*) and 1 *Gsx* ParaHox genes. Compared to hexapods, we identified expansions in some NK-like genes, including 5 Bar homeobox genes (*BarH1/2*), 2 developing brain homeobox genes (*DBX*) and 6 muscle segment homeobox genes (*MSX/Drop*). Evidence from several bilaterian genomes suggests that *NK* genes are clustered together (*Pollard and Holland, 2000*; *Jagla et al., 2001*; *Luke et al., 2003*; *Castro and Holland, 2003*). In the current assembly of the *Parhyale* genome, we identified an *NK2-3* gene and an *NK3* gene on the same scaffold (phaw_30.0004720) and the tandem duplication of an *NK2* gene on another scaffold (phaw_30.0004663). Within the *ANTP* class, we also observed 1 mesenchyme homeobox (*Meox*), 1 motor neuron homeobox (*MNX/Exex*) and 3 even-skipped homeobox (*Evx*) genes.

## The *Parhyale* genome encodes glycosyl hydrolase enzymes consistent with lignocellulose digestion ('wood eating')

Lignocellulosic (plant) biomass is the most abundant raw material on our planet and holds great promise as a source for the production of bio-fuels (*Himmel et al., 2007*). Understanding how some animals and their symbionts achieve lignocellulose digestion is a promising research avenue for exploiting lignocellulose-rich material (*Wilson, 2011*; *Cragg et al., 2015*). Amongst Metazoans, research into the ability to depolymerize plant biomass into useful catabolites is largely restricted to terrestrial species such as ruminants, termites and beetles. These animals rely on mutualistic associations with microbial endosymbionts that provide cellulolytic enzymes known as glycosyl hydrolases (GHs) (*Duan et al., 2009*; *Warnecke et al., 2007*) (*Figure 11*). Much less studied is lignocellulose digestion in aquatic animals despite the fact that lignocellulose represents a major energy source in aquatic environments, particularly for benthic invertebrates (*Distel et al., 2011*). Recently, it has been suggested that the marine wood-boring Isopod *Limnoria quadripunctata* and the amphipod *Chelura terebrans* may have sterile microbe-free digestive systems and they produce all required enzymes for lignocellulose digestion (*King et al., 2010*; *Green Etxabe, 2013*; *Kern et al., 2013*). Significantly, these species have been shown to have endogenous GH7 family enzymes with cellobio-hydrolase (beta-1,4-exoglucanase) activity, previously thought to be absent from animal genomes. From an evolutionary perspective, it is likely that GH7 coding genes were acquired by these species via horizontal gene transfer from a protist symbiont.

*Parhyale* is a detrivore that can be sustained on a diet of carrots (*Figure 11C*), suggesting that they too may be able to depolymerize lignocellulose for energy (*Figure 11A and B*). We searched for GH family genes in *Parhyale* using the classification system of the CAZy (Carbohydrate-Active enZYmes) database (*Cantarel et al., 2009*) and the annotation of protein domains in predicted genes with PFAM (*Finn et al., 2006*). We identified 73 GH genes with complete GH catalytic domains that were classified into 17 families (*Figure 12—source data 1*) including 3 members of the GH7 family. Phylogenetic analysis of *Parhyale* GH7s show high sequence similarity to the known GH7 genes in *L. quadripunctata* and the amphipod *C. terebrans* (*Kern et al., 2013*) (*Figure 12A*; *Figure 12—figure supplement 1*). GH7 family genes were also identified in the transcriptomes of three more species spanning the multicrustacea clade: *Echinogammarus veneris* (amphipod), *Eucyclops serrulatus* (copepod) and *Calanus finmarchicus* (copepod). As previously reported, we also discovered a closely related GH7 gene in the branchiopod *Daphnia* (*Figure 12A*) (*Cragg et al., 2015*). This finding supports the grouping of Branchiopoda with Multicrustacea (rather than with Hexapoda) and the acquisition of a GH7 gene by a vericrustacean ancestor. Alternatively, this suggests an even earlier acquisition of a GH7 gene by a crustacean ancestor with subsequent loss of the GH7 family gene in the lineage leading to insects.

GH families 5, 9, 10, and 45 encode beta-1,4-endoglucanases which are also required for lignocellulose digestion and are commonly found across Metazoa. We found 3 GH9 family genes with complete catalytic domains in the *Parhyale* genome as well as in the other three multicrustacean species (*Figure 12B*). These GH9 enzymes exhibited a high sequence similarity to their homologues in the isopod *Limnoria* and in a number of termites. Beta-glucosidases are the third class of enzyme required for digestion of lignocellulose. They have been classified into a number of GH families: 1, 3, 5, 9 and 30, with GH1 representing the largest group (*Cantarel et al., 2009*). In *Parhyale*, we found 7 beta-glucosidases from the GH30 family and 3 from the GH9 family, but none from the GH1 family.

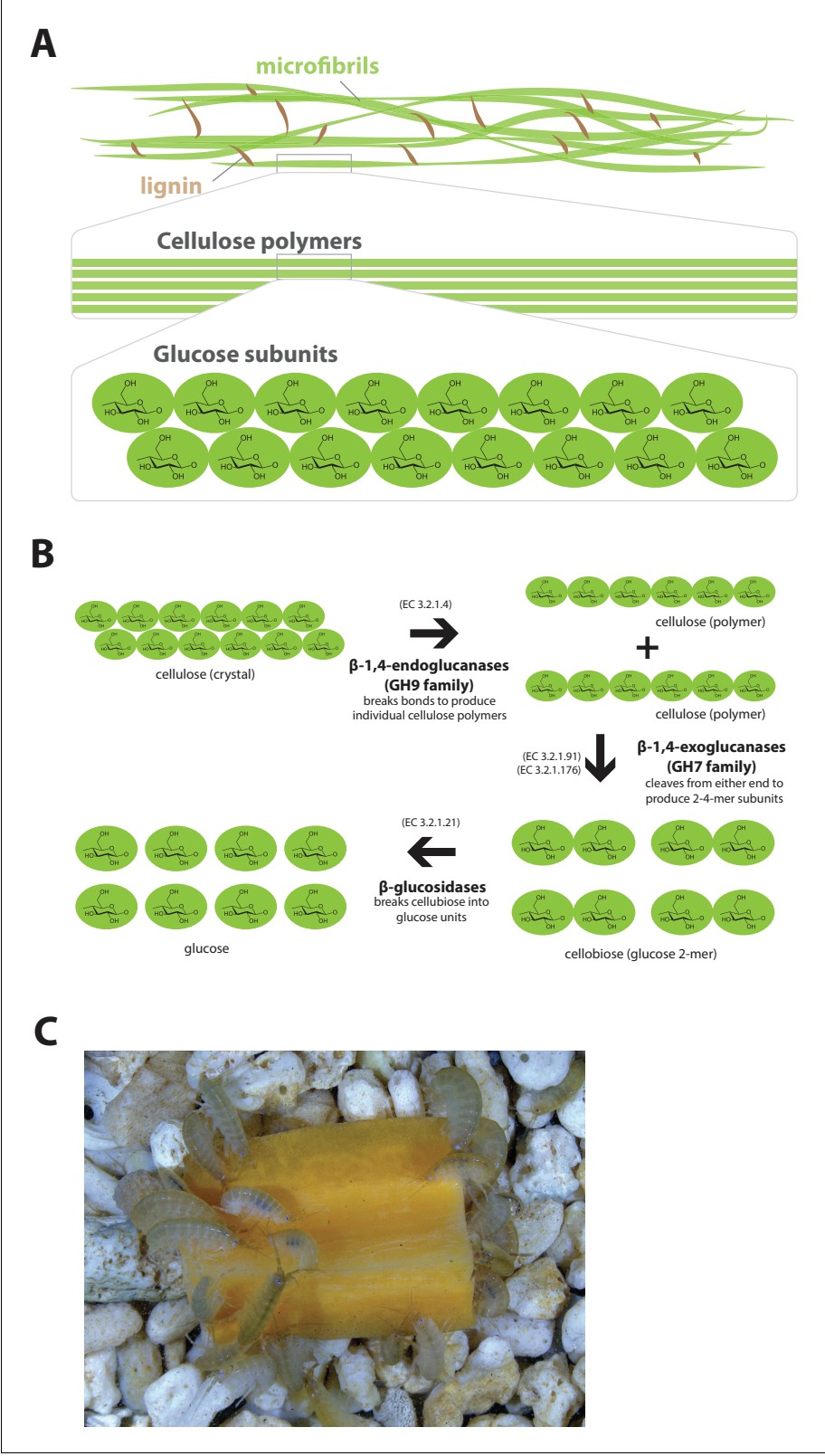

**Figure 11.** Lignocellulose digestion overview. (**A**) Simplified drawing of lignocellulose structure. The main component of lignocellulose is cellulose, which is a-1,4-linked chain of glucose monosaccharides. Cellulose and lignin are organized in structures called microfibrils, which in turn form macrofibrils. (**B**) Summary of cellulolytic enzymes and reactions involved in the breakdown of cellulose into glucose. -1,4-endoclucanases of the GH9
*Figure 11 continued on next page*

*Figure 11 continued*

family catalyze the hydrolysis of crystalline cellulose into cellulose chains. -1,4-exoclucanases of the GH7 family break down cellulose chains into cellobiose (glucose disaccharide) that can be converted to glucose by -glucosidases. (C) Adult *Parhyale* feeding on a slice of carrot.

Understanding lignocellulose digestion in animals using complex mutualistic interactions with microbes has proven to be a difficult task. The study of 'wood-eating' in *Parhyale* can offer new insights into lignocellulose digestion in the absence of gut microbes, and the unique opportunity to apply molecular genetic approaches to understand the activity of glycosyl hydrolases in the digestive system. Lignocellulose digestion may also have implications for gut immunity in some crustaceans, since these reactions have been reported to take place in a sterile gut (*Boyle and Mitchell, 1978*; *Zimmer et al., 2002*).

## Characterisation of the innate immune system in a Malacostracan

Immunity research in Malacostracans has attracted interest due to the rapid rise in aquaculture related problems (*Vazquez et al., 2009*; *Stentiford et al., 2012*; *Hauton, 2012*). Malacostracan food crops represent a huge global industry (>$40 Billion at point of first sale), and reliance on this crop as a source of animal protein is likely to increase in line with human population growth (*Stentiford et al., 2012*). Here we provide an overview of immune-related genes in *Parhyale* that were identified by mapping proteins to the ImmunoDB database (*Waterhouse et al., 2007*). The ability of the innate immune system to identify pathogen-derived molecules is mediated by pattern recognition receptors (PRRs) (*Janeway and Medzhitov, 2002*). Several groups of invertebrate PRRs have been characterized, i.e. thioester-containing proteins (*TEP*), Toll-like receptors (*TLR*), peptido-glycan recognition proteins (*PGRP*), C-type lectins, galectins, fibrinogen-related proteins (*FREP*),

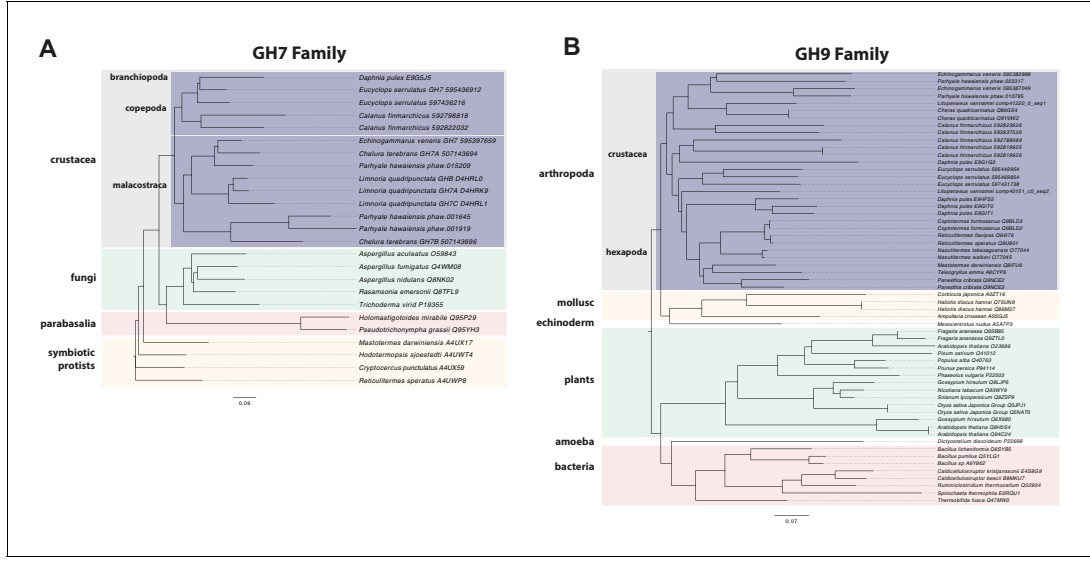

**Figure 12.** Phylogenetic analysis of GH7 and GH9 family proteins. (A) Phylogenetic tree showing the relationship between GH7 family proteins of *Parhyale*, other crustaceans (Malacostraca, Branchiopoda, Copepoda), fungi and symbiotic protists (root). UniProt and GenBank accessions are listed next to the species names. (B) Phylogenetic tree showing the relationship between GH9 family proteins of *Parhyale*, crustaceans, insects, molluscs, echinoderms, amoeba, bacteria and plants (root). UniProt and GenBank accessions are listed next to the species names. Both trees were constructed with RAxML using the WAG+G model from multiple alignments of protein sequences created with MUSCLE.

The following source data and figure supplement are available for figure 12:

**Source data 1.** Catalog of GH family genes in *Parhyale*.

**Figure supplement 1.** Alignment of GH7 family genes.

gram-negative binding proteins (*GNBP*), Down Syndrome Cell Adhesion Molecules (*Dscam*) and lip-opolysaccharides and beta-1, 3-glucan binding proteins (*LGBP*).

The functions of *PGRPs* have been described in detail in insects like *D. melanogaster* (*Werner et al., 2003*) and the PGRP family has also been reported in Vertebrates, Molluscs and Echinoderms (*Liu et al., 2001*; *Rehman et al., 2001*). Surprisingly, we found no PGRP genes in the *Parhyale* genome. *PGRPs* were also not found in other sequence datasets from Branchiopoda, Copepoda and Malacostraca (*Figure 13A*), raising the possibility of their close phylogenetic relationship (like the GH7 genes). In the absence of *PGRPs*, the freshwater crayfish *Pacifastacus leniusculus* relies on a Lysine-type peptidoglycan and serine proteinases, *SPH1* and *SPH2* that forms a complex with *LGBP* during immune response (*Liu et al., 2011*). In *Parhyale*, we found one LGBP gene and two serine proteinases with high sequence identity to *SPH1/2* in *Pacifastacus*. The *D. pulex* genome has also an expanded set of Gram-negative binding proteins (proteins similar to *LGBP*) suggesting a compensatory mechanism for the lost *PGRPs* (*McTaggart et al., 2009*). Interestingly, we found a putative *PGRP* in the Remipede *Speleonectes tulumensis* (*Figure 13A*) providing further support for sister group relationship of Remipedia and Hexapoda (*von Reumont et al., 2012*).

Innate immunity in insects is transduced by three major signaling pathways: the Immune Deficiency (*Imd*), Toll and Janus kinase/signal transducer and activator of transcription (*JAK/STAT*) pathways (*Dostert et al., 2005*; *Tanji et al., 2007*). We found 16 members of the Toll family in *Parhyale* including 10 Toll-like receptors (TLRs) (*Figure 13B*). Some TLRs have been also implicated in embryonic tissue morphogenesis in *Parhyale* and other arthropods (*Benton et al., 2016*). Additionally, we identified 7 Imd and 25 JAK/STAT pathway members including two negative regulators: suppressor of cytokine signaling (*SOCS*), and protein inhibitor of activated *STAT* (*PIAS*) (*Arbouzova and Zeidler, 2006*) (*Figure 13—source data 1*).

The blood of arthropods (hemolymph) contains hemocyanin which is a copper-binding protein involved in the transport of oxygen, and circulating blood cells called hemocytes for the phagocytosis of pathogens. Phagocytosis by hemocytes is facilitated by the evolutionarily conserved gene family, the thioester-containing proteins (*TEPs*) (*Levashina et al., 2001*). Previously sequenced Pancrustacean species contained between 2 to 52 *TEPs*. We find 5 *TEPs* in the *Parhyale* genome. Arthropod hemocyanins themselves are structurally related to phenoloxidases (PO; (*Decker and Jaenicke, 2004*) and can be converted into POs by conformational changes under specific conditions (*Lee et al., 2004*). POs are involved in several biological processes (like the melanization immune response, wound healing and cuticle sclerotization) and we identified 7 PO genes in *Parhyale*. Interestingly, hemocyanins and PO activity have been shown to be highly abundant together with glycosyl hydrolases in the digestive system of Isopods and Amphipods, raising a potential mechanistic link between gut sterility and degradation of lignocellulose (*King et al., 2010*; *Zimmer et al., 2002*).

Another well-studied transmembrane protein essential for neuronal wiring and adaptive immune responses in insects is the immunoglobulin (*Ig*)-superfamily receptor Down syndrome cell adhesion molecule (*Dscam*) (*Schmucker et al., 2000*; *Watson et al., 2005*). Alternative splicing of *Dscam* transcripts can result in thousands of different isoforms that have a common architecture but have sequence variations encoded by blocks of alternative spliced exons. The *D. melanogaster Dscam* locus encodes 12 alternative forms of exon 4 (encoding the N-terminal half of Ig2), 48 alternative forms of exon 6 (encoding the N-terminal half of Ig3), 33 alternative forms of exon 9 (encoding Ig7), and 2 alternative forms of exon 17 (encoding transmembrane domains) resulting in a total of 38,016 possible combinations. The *Dscam* locus in *Parhyale* (and in other crustaceans analysed) has a similar organization to insects; tandem arrays of multiple exons encode the N-terminal halves of Ig2 (exon 4 array with at least 13 variants) and Ig3 (exon 6 array with at least 20 variants) and the entire Ig7 domain (exon 14 array with at least 13 variants) resulting in at least 3380 possible combinations (*Figure 13C–E*). The alternative splicing of hypervariable exons in *Parhyale* was confirmed by sequencing of cDNA clones amplified with Dscam-specific primers. Almost the entire *Dscam* gene is represented in a single genomic scaffold and exhibits high amino-acid sequence conservation with other crustacean *Dscams* (*Figure 13——figure supplement 1*). The number of *Dscam* isoforms predicted in *Parhyale* is similar to that predicted for Daphnia species (*Brites et al., 2008*). It remains an open question whether the higher number of isoforms observed in insects coincides with the evolution of additional Dscam functions compared to crustaceans.

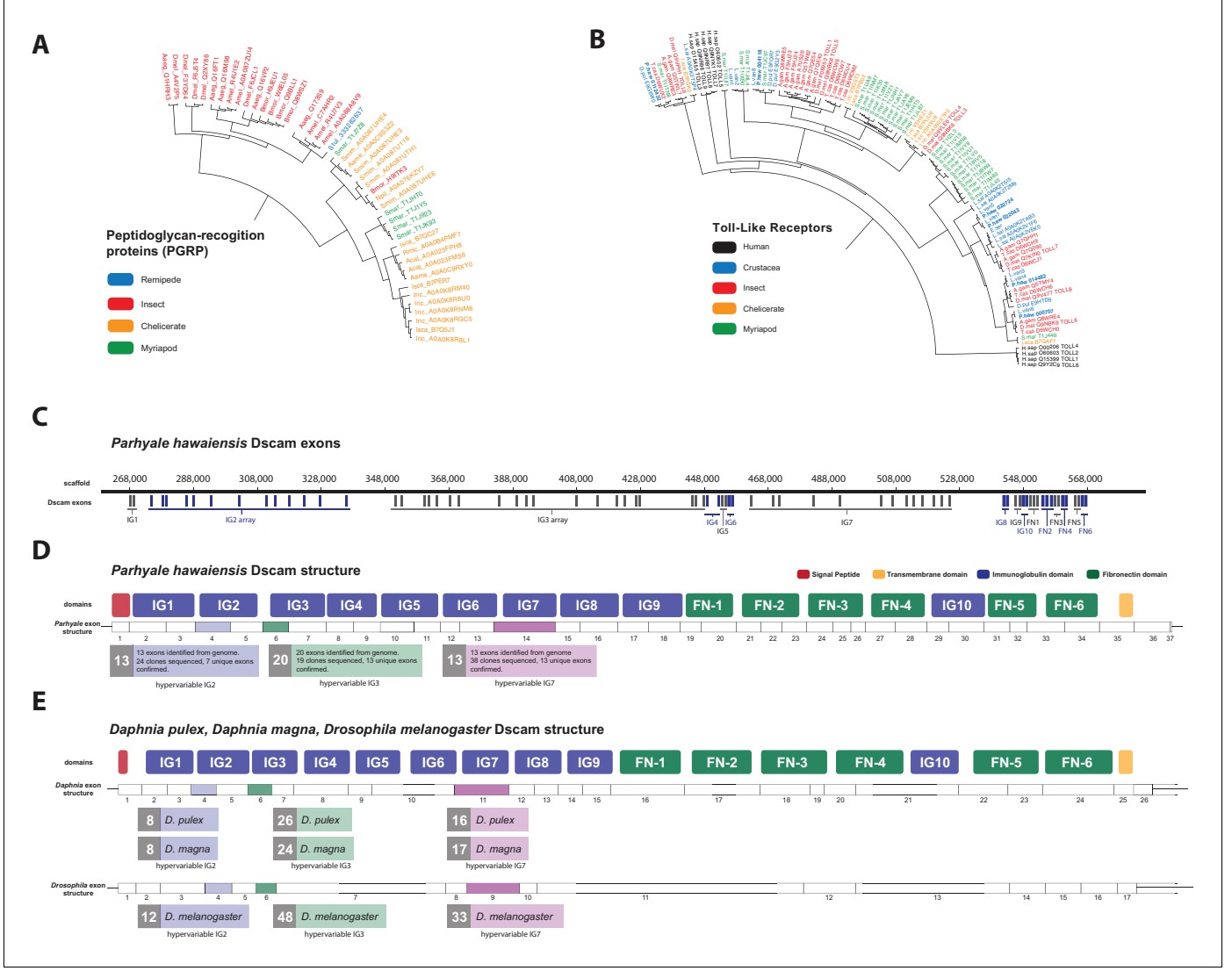

**Figure 13.** Comparison of innate immunity genes. (**A**) Phylogenetic tree of peptidoglycan recognition proteins (PGRPs). With the exception of Remipedes, PGRPs were not found in Crustaceans. PGRPs have been found in Arthropods, including insects, Myriapods and Chelicerates. (**B**) Phylogenetic tree of Toll-like receptors (TLRs) generated from five Crustaceans, three Hexapods, two Chelicerates, one Myriapod and one vertebrate species. (**C**) Genomic organization of the *Parhyale* Dscam locus showing the individual exons and exon arrays encoding the immunoglobulin (IG) and fibronectin (FN) domains of the protein. (**D**) Structure of the *Parhyale* Dscam locus and comparison with the (**E**) Dscam loci from *Daphnia pulex, Daphnia magna* and *Drosophila melanogaster*. The white boxes represent the number of predicted exons in each species encoding the signal peptide (red), the IGs (blue), the FNs and transmembrane (yellow) domains of the protein. The number of alternatively spliced exons in the arrays encoding the hypervariable regions IG2 (exon 4 in all species), IG3 (exon 6 in all species) and IG7 (exon 14 in *Parhyale*, 11 in *D. pulex* and 9 in *Drosophila*) are indicated under each species schematic in the purple, green and magenta boxes, respectively. Abbreviations of species used: *Parhyale hawaiensis* (Phaw), *Bombyx mori* (Bmor), *Aedes aegypti* (Aaeg), *Drosophila melanogaster* (Dmel), *Apis mellifera* (Amel), *Speleonectes tulumensis* (Stul), *Strigamia maritima* (Smar), *Stegodyphus mimosarum* (Smim), *Ixodes scapularis* (Isca), *Amblyomma americanum* (Aame), *Nephila pilipes* (Npil), *Rhipicephalus microplus* (Rmic), *Ixodes ricinus* (Iric), *Amblyomma cajennense* (Acaj), *Anopheles gambiae* (Agam), *Daphnia pulex* (Apul), *Tribolium castaneum* (Tcas), *Litopenaeus vannamei* (Lvan), *Lepeophtheirus salmonis* (Lsal), *Eucyclops serrulatus* (Eser), *Homo sapiens* (H.sap). Both trees were constructed with RAxML using the WAG+G model from multiple alignments of protein sequences created with MUSCLE.

The following source data and figure supplement are available for figure 13:

**Source data 1.** Catalog of innate immunity related genes in *Parhyale*.

**Figure supplement 1.** Overview of *Parhyale* Dscam structure and hypervariable regions

From a functional genomics perspective, the *Parhyale* immune system appears to be a good representative of the malacostrocan or even multicrustacean clade that can be studied in detail with existing tools and resources.

## Non-coding RNAs and associated proteins in the *Parhyale* genome

Non-coding RNAs are a central, but still a relatively poorly understood part of eukaryotic genomes. In animal genomes, different classes of small RNAs are key for genome surveillance, host defense against viruses and parasitic elements in the genome, and regulation of gene expression through transcriptional, post-transcriptional and epigenetic control mechanisms (*Castel and Martienssen, 2013*; *Aravin et al., 2001*; *Caplen et al., 2001*; *Brennecke et al., 2007*; *Gu et al., 2009*; *Lee et al., 2012*; *He and Hannon, 2004*; *Thomson et al., 2006*; *Filipowicz et al., 2008*). The nature of these non-coding RNAs, as well as the proteins involved in their biogenesis and function, can vary between animals. For example, some nematodes have Piwi-interacting short RNAs (piRNAs), while others have replaced these by alternate small RNA based mechanisms to compensate for their loss (*Sarkies et al., 2015*).

As a first step, we surveyed the *Parhyale* genome for known conserved protein components of the small interfering RNA (siRNA/RNAi) and the piRNA pathways (*Table 4*). We found key components of all major small RNA pathways, including 4 argonaute family members, 2 PIWI family members, and orthologs of *D. melanogaster Dicer-1* and *Dicer-2*, *drosha* and *loquacious*, (*Figure 14—figure supplement 1*). Among Argonaute genes, *Parhyale* has 1 *AGO-1* ortholog and 3 *AGO-2* orthologs, which is presumably a malacostraca-specific expansion. While *Parhyale* only has 2 PIWI family members, other crustacean lineages have clearly undergone independent expansions of this protein family. Unlike in *C. elegans*, many mammals, fish and insects (but not *D. melanogaster*), we did not find any evidence in the *Parhyale* genome for the *SID-1* (systemic RNA interference defective) transmembrane protein that is essential for systemic RNAi (*Dong and Friedrich, 2005*; *Honeybee Genome Sequencing Consortium, 2006*; *Xu and Han, 2008*). Species without a *SID-1*

**Table 4.** Small RNA processing pathway members. The *Parhyale* orthologs of small RNA processing pathway members.

| Gene | Counts | Gen ID |
|---|---|---|
| Armitage | 2 | phaw_30_tra_m.006391<br>phaw_30_tra_m.007425 |
| Spindle_E | 3 | phaw_30_tra_m.000091<br>phaw_30_tra_m.020806<br>phaw_30_tra_m.018110 |
| rm62 | 7 | phaw_30_tra_m.014329<br>phaw_30_tra_m.012297<br>phaw_30_tra_m.004444<br>phaw_30_tra_m.012605<br>phaw_30_tra_m.001849<br>phaw_30_tra_m.006468<br>phaw_30_tra_m.023485 |
| Piwi/<br>aubergine | 2 | phaw_30_tra_m.011247<br>phaw_30_tra_m.016012 |
| Dicer 1 | 1 | phaw_30_tra_m.001257 |
| Dicer 2 | 1 | phaw_30_tra_m.021619 |
| argonaute 1 | 1 | phaw_30_tra_m.006642 |
| arogonaute 2 | 3 | phaw_30_tra_m.021514<br>phaw_30_tra_m.018276<br>phaw_30_tra_m.012367 |
| Loquacious | 2 | phaw_30_tra_m.006389<br>phaw_30_tra_m.000074 |
| Drosha | 1 | phaw_30_tra_m.015433 |

ortholog can silence genes only in a cell-autonomous manner (*Roignant et al., 2003*). This feature has important implications for future design of RNAi experiments in *Parhyale*.

We also assessed the miRNA and putative long non-coding RNAs (lncRNA) content of *Parhyale* using both MiRPara and Rfam (*Wu et al., 2011*; *Nawrocki et al., 2015*). We annotated 1405 homologues of known non-coding RNAs using Rfam. This includes 980 predicted tRNAs, 45 rRNA of the large ribosomal subunit, 10 rRNA of the small ribosomal subunit, 175 snRNA components of the major spliceosome (U1, U2, U4, U5 and U6), 5 snRNA components of the minor spliceosome (U11, U12, U4atac and U6atac), 43 ribozymes, 38 snoRNAs, 71 conserved cis-regulatory element derived RNAs and 42 highly conserved miRNA genes ( *Source code 6*). *Parhyale* long non-coding RNAs (lncRNAs) were identified from the transcriptome using a series of filters to remove coding transcripts producing a list of 220,284 putative lncRNAs (32,223 of which are multi-exonic). Only one *Parhyale* lncRNA has clear homology to another annotated lncRNA, the sphinx lncRNA from *D. melanogaster* (*Wang et al., 2002*).

We then performed a more exhaustive search for miRNAs using MiRPara (*Source code 6*) and a previously published *Parhyale* small RNA read dataset (*Blythe et al., 2012*). We identified 1403 potential miRNA precursors represented by 100 or more reads. Combining MiRPara and Rfam results, we annotated 31 out of the 34 miRNA families found in all Bilateria, 12 miRNAs specific to Protostomia, 4 miRNAs specific to Arthropoda and 5 miRNAs previously found to be specific to Mandibulata (*Figure 14*). We did not identify *mir-125, mir-283* and *mir-1993* in the *Parhyale* genome. The absence of *mir-1993* is consistent with reports that this miRNA was lost during Arthropod evolution (*Wheeler et al., 2009*). While we did not identify *mir-125*, we observed that *mir-100* and *let-7* occurred in a cluster on the same scaffold (*Figure 14—figure supplement 2*), where *mir-125* is also present in other animals. The absence of *mir-125* has been also reported for the centipede genome (*Chipman et al., 2014*). *mir-100* is one of the most primitive miRNAs shared by Bilateria and Cnidaria (*Grimson et al., 2008*; *Wheeler et al., 2009*). The distance between *mir-100* and *let-7* genes within the cluster can vary substantially between different species. Both genes in *Parhyale* are localized within a 9.3kb region (*Figure 14—figure supplement 2*) as compared to 3.8kb in the mosquito *Anopheles gambiae* and 100bp in the beetle *Tribolium* (*Behura, 2007*). Similar to *D. melanogaster* and the polychaete *Platynereis dumerilii*, we found that *Parhyale mir-100* and let-7 are co-transcribed as a single, polycistronic lncRNA. We also found another cluster with *miR-71* and *mir-2* family members which is conserved across many invertebrates (*Marco et al., 2014*) (*Figure 14—figure supplement 2*).

Conserved linkages have also been observed between miRNAs and Hox genes in Bilateria (*Enright et al., 2003a*; *Tanzer et al., 2005*; *Lemons and McGinnis, 2006*; *Stark et al., 2008*; *Shippy et al., 2008*). For example, the phylogenetically conserved *mir-10* is present within both vertebrate and invertebrate Hox clusters between Hoxb4/*Dfd* and *Hoxb5/Scr* (*Enright et al., 2003b*). In the *Parhyale* genome and Hox BAC sequences, we found that *mir-10* is also located between *Dfd* and *Src* on BAC clone PA179-K23 and scaffold phaw_30.0001203 (*Figure 14——figure supplement 2*). However, we could not detect *mir-iab-4* near the *Ubx* and *AbdA* genes in *Parhyale*, the location where it is found in other arthropods/insects (*Cumberledge et al., 1990*).

Preliminary evidence regarding the presence of PIWI proteins and other piRNA pathway proteins also suggests that the piRNA pathway is likely active in *Parhyale*, although piRNAs themselves await to be surveyed. The opportunity to study these piRNA, miRNA and siRNA pathways in a genetically tractable crustacean system will shed further light into the regulation and evolution of these pathways and their contribution to morphological diversity.

## Methylome analysis of the *Parhyale* genome

Methylation of cytosine residues (m5C) in CpG dinucleotides in animal genomes is regulated by a conserved multi-family group of DNA methyltransferases (DNMTs) with diverse roles in the epigenetic control of gene expression, genome stability and chromosome dynamics (*Zemach et al., 2010*; *Law and Jacobsen, 2010*; *Jones, 2012*). The phylogenetic distribution of DNMTs in Metazoa suggests that the bilaterian ancestor had at least one member of the Dnmt1 and Dnmt3 families (involved in de novo methylation and maintenance of DNA methylation) and the Dnmt2 family (involved in tRNA methylation), as well as additional RNA methyltransferases (*Jones and Liang, 2009*; *Jeltsch et al., 2016*). Many animal groups have lost some of these DNA methyltransferases, for example *DNMT1* and 3 are absent from *D. melanogaster* and flatworms (*Goll et al., 2006*;

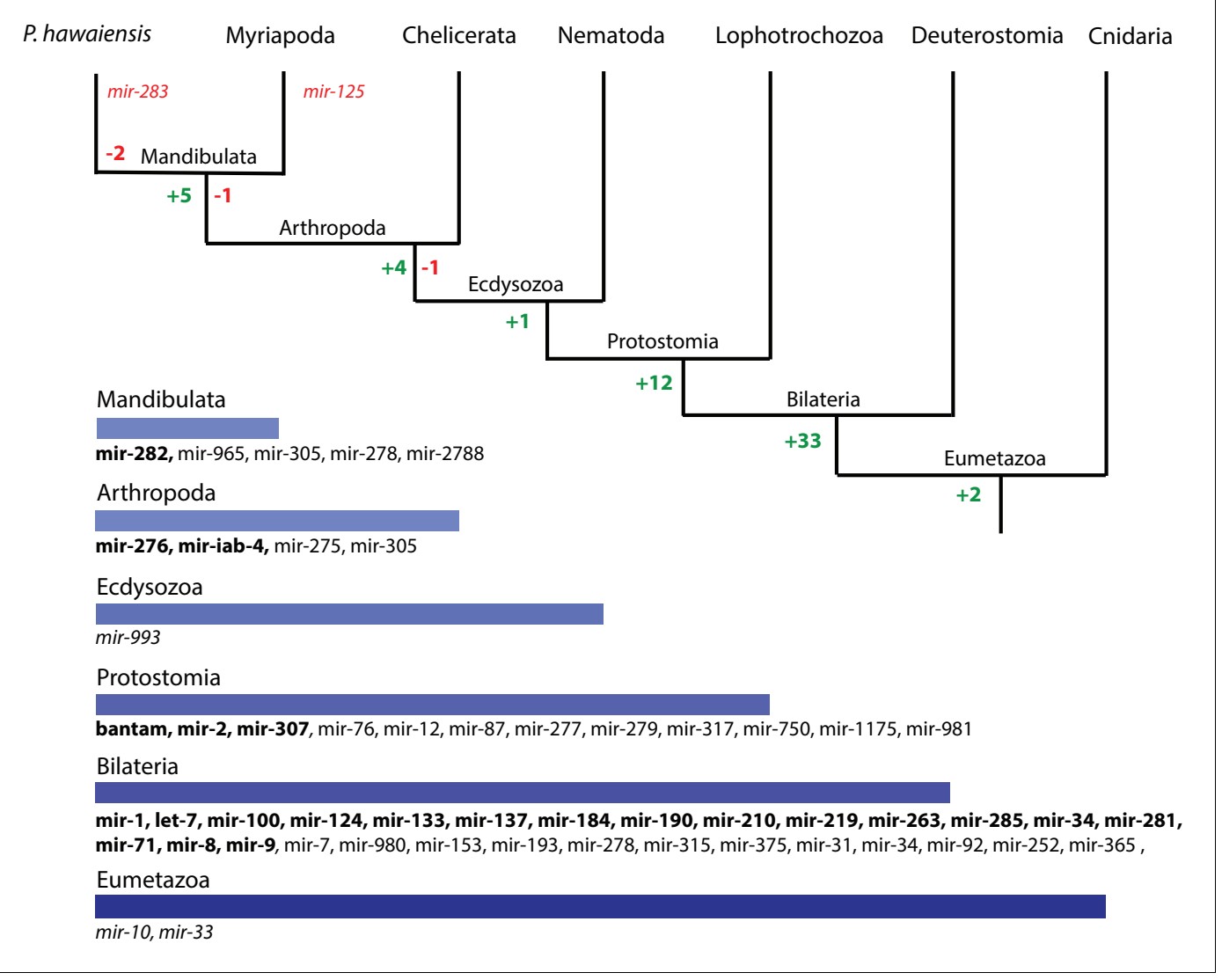

**Figure 14.** Evolution of miRNA families   in Eumetazoans.   Phylogenetic tree showing the gains (in green) and losses (in red) of miRNA families at various taxonomic levels of the Eumetazoan tree leading to *Parhyale*. miRNAs marked with plain characters were identified by MirPara with small RNA sequencing read support. miRNAs marked with bold characters were identified by Rfam and MirPara with small RNA sequencing read support.

The following source data and figure supplements are available for figure 14:

**Source data 1.** RFAM based annotation of the *Parhyale* genome.

**Figure supplement 1.** Phylogenetic trees of Dicer and PIWI/AGO genes.

**Figure supplement 2.** Examples of miRNAs in the *Parhyale* genome.

*Jaber-Hijazi et al., 2013*), while *DNMT2* is absent from nematodes *C. elegans* and *C. briggsae*. The *Parhyale* genome encodes members of all 3 families *DNMT1, DNMT3* and *DNMT2*, as well as 2 orthologs of conserved methyl-CpG-binding proteins and a single orthologue of *Tet2*, an enzyme involved in DNA demethylation (*Hackett et al., 2013*) (*Figure 15A* and *Figure 15—source data 1*).

We used genome wide bisulfite sequencing to confirm the presence and also assess the distribution of CpG dinucleotide methylation. Our results indicated that 20–30% of *Parhyale* DNA is methylated at CpG dinucleotides (*Figure 15B*). The *Parhyale* methylation pattern is similar to that

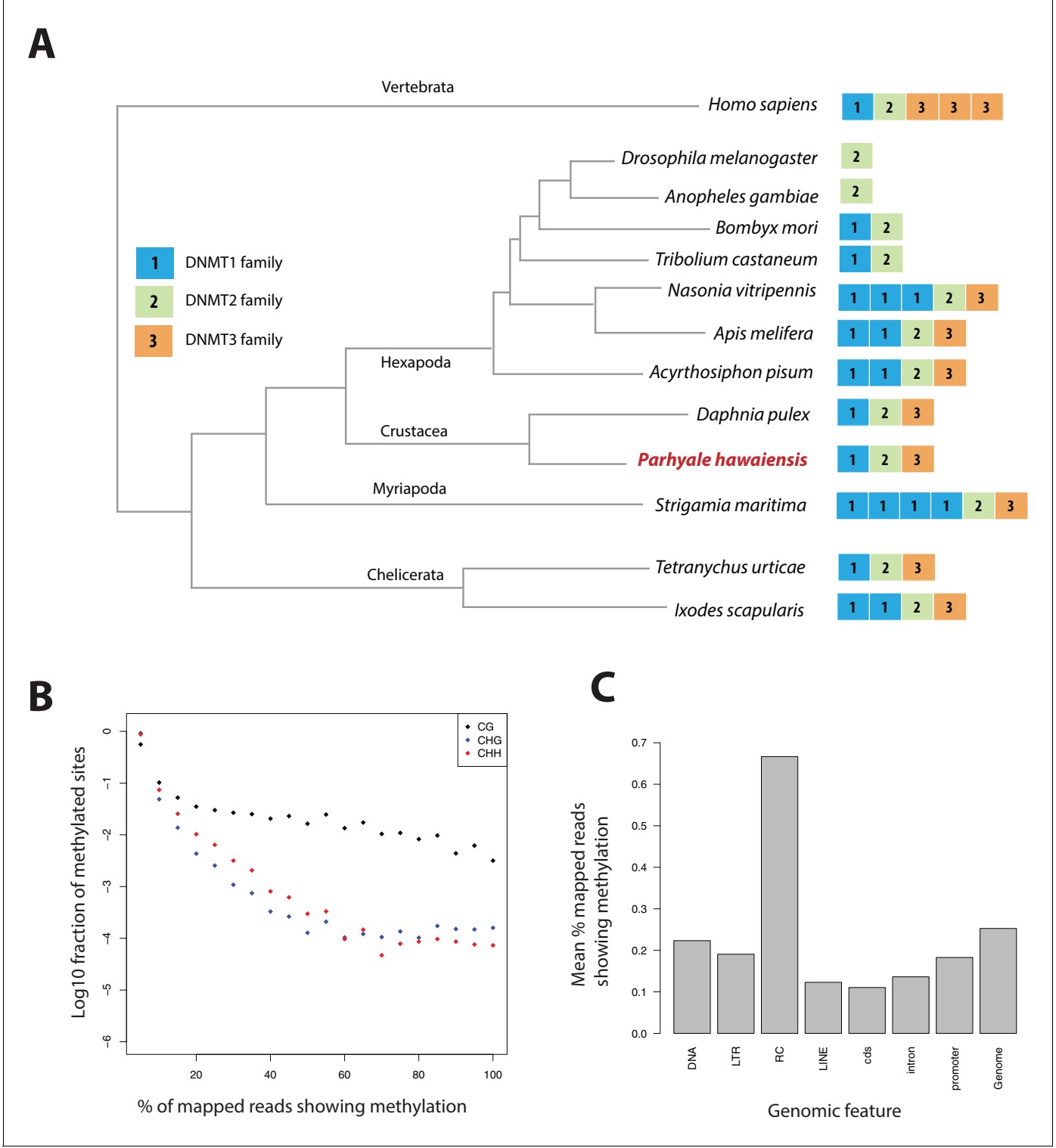

**Figure 15.** Analysis of Parhyale genome methylation. (**A**) Phylogenetic tree showing the families and numbers of DNA methyltransferases (DNMTs) present in the genomes of indicated species. *Parhyale* has one copy from each DNMT family. (**B**) Amounts of methylation detected in the *Parhyale* genome. Amount of methylation is presented as percentage of reads showing methylation in bisulfite sequencing data. DNA methylation was analyzed in all sequence contexts (CG shown in dark, CHG in blue and CHH in red) and was detected preferentially in CpG sites. (**C**) Histograms showing mean percentages of methylation in different fractions of the genome: DNA transposons (DNA), long terminal repeat transposable elements (LTR), rolling circle transposable elements (RC), long interspersed elements (LINE), coding sequences (cds), introns, promoters, and the rest of the genome.

*Figure 15 continued on next page*

*Figure 15 continued*

The following source data is available for figure 15:

**Source data 1.** Genes involved with epigenetic modification.

observed in vertebrates, with high levels of methylation detected in transposable elements and other repetitive elements, in promoters and gene bodies (*Figure 15C*). A particular class of rolling-circle transposons are very highly methylated in the genome, potentially implicating methylation in silencing these elements. For comparison, about 1% or less of CpG-associated cytosines are methylated in insects like *Drosophila, Apis, Bombyx* and *Tribolium*. (*Feng et al., 2010*; *Jeltsch, 2010*; *Zemach et al., 2010*). These data represent the first documentation of a crustacean methylome. Considering the utility of *Parhyale* for genetic and genomic research, we anticipate future investigations to shed light on the functional importance and spatiotemporal dynamics of epigenetic modifications during normal development and regeneration, as well as their relevance to equivalent processes in vertebrate systems.

## *Parhyale* genome editing using homology-independent approaches

*Parhyale* has already emerged as a powerful model for developmental genetic research where the expression and function of genes can be studied in the context of stereotyped cellular processes and with a single-cell resolution. Several experimental approaches and standardized resources have been established to study coding and non-coding sequences (*Table 1*). These functional studies will be enhanced by the availability of the assembled and annotated genome presented here. As a first application of these resources, we tested the efficiency of the CRISPR/Cas system for targeted genome editing in *Parhyale* (*Mali et al., 2013*; *Jinek et al., 2012*; *Cong et al., 2013*; *Gilles and Averof, 2014*; *Martin et al., 2015*; *Serano et al., 2015*). In these studies, we targeted the *Distal-less* patterning gene (called *PhDll-e*) (*Liubicich et al., 2009*) that has a widely-conserved and highly-specific role in animal limb development (*Panganiban et al., 1997*).

We first genotyped our wild-type laboratory culture and found two *PhDll-e* alleles with 23 SNPs and 1 indel in their coding sequences and untranslated regions. For *PhDll-e* knock-out, two sgRNAs targeting both alleles in their coding sequences downstream of the start codon and upstream of the DNA-binding homeodomain were injected individually into 1-cell-stage embryos (G0 generation) together with a transient source of Cas9 (*Figure 16—figure supplement 1 A-B*). Both sgRNAs gave rise to animals with truncated limbs (*Figure 16A and B*); the first sgRNA at a relatively low percentage around 9% and the second one at very high frequencies ranging between 53% and 76% (*Figure 16—figure supplement 1*). Genotyping experiments revealed that injected embryos carried *PhDll-e* alleles modified at the site targeted by each sgRNA (*Figure 16—figure supplement 1B*). The number of modified *PhDll-e* alleles recovered from G0s varied from two, in cases of early bi-allelic editing at the 1-cell-stage, to three or more, in cases of later-stage modifications by Cas9 (*Figure 16—figure supplement 1C*). We isolated indels of varying length that were either disrupting the open reading frame, likely producing loss-of-function alleles or were introducing in-frame mutations potentially representing functional alleles (*Figure 16—figure supplement 1C–D*). In one experiment with the most efficient sgRNA, we raised the injected animals to adulthood and set pairwise crosses between 17 fertile G0s (10 male and 7 female): 88% (15/17) of these founders gave rise to G1 offspring with truncated limbs, presumably by transmitting *PhDll-e* alleles modified by Cas9 in their germlines. We tested this by genotyping individual G1s from two of these crosses and found that embryos bearing truncated limbs were homozygous for loss-of-function alleles with out-of-frame deletions, while their wild-type siblings carried one loss-of-function allele and one functional allele with an in-frame deletion (*Figure 16—figure supplement 1 D*).

The non-homologous end joining (NHEJ) repair mechanism operating in the injected cells can be exploited not only for gene knock-out experiments described above, but also for CRISPR knock-in approaches where an exogenous DNA molecule is inserted into the targeted locus in a homology-independent manner. This homology-independent approach could be particularly useful for *Parhyale* that exhibits high levels of heterozygosity and polymorphisms in the targeted laboratory populations, especially in introns and intergenic regions. To this end, we co-injected into 1-cell-stage

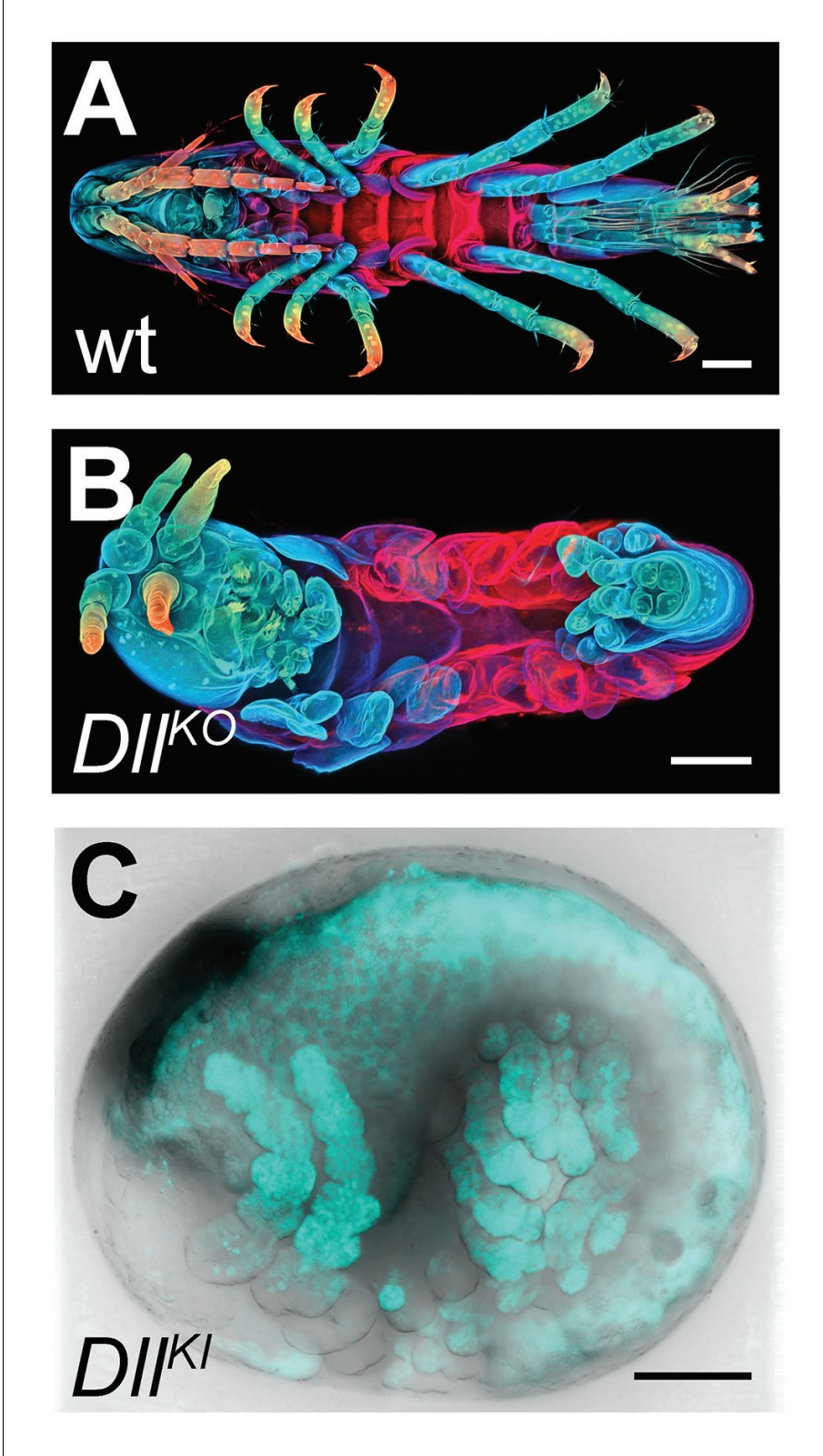

**Figure 16.** CRISPR/Cas9-based genome editing in Parhyale. (**A**) Wild-type morphology. (**B**) Mutant *Parhyale* with truncated limbs after CRISPR-mediated knock-out (DllKO) of the limb patterning gene *Distal-less* (*PhDll-e*). Panels show ventral views of juveniles stained for cuticle and color-coded by depth with anterior to the left. (**C**) Fluorescent tagging of *PhDll-e* expressed in most limbs (shown in cyan) by CRISPR-mediated knock-in (DllKI) using

*Figure 16 continued on next page*

*Figure 16 continued*

the non-homologous-end-joining repair mechanism. Panel shows a lateral view with anterior to the left and dorsal to the top of a live embryo (stage S22) with merged bright-field and fluorescence channels. Yolk autofluorescence produces a dorsal crescent of fluorescence in the gut. Scale bars are 100 μm.

The following figure supplement is available for figure 16:

**Figure supplement 1.** CRISPR experiments targeting the Distalless locus.

---

embryos the Cas9 protein together with the strongest sgRNA and a tagging plasmid. The plasmid was designed in such a way that upon its linearization by the same sgRNA and Cas9 and its integration into the *PhDll-e* locus in the appropriate orientation and open reading frame, it would restore the endogenous *PhDll-e* coding sequence in a bicistronic mRNA also expressing a nuclear fluorescent reporter. Among injected G0s, about 7% exhibited a nuclear fluorescence signal in the distal (telopodite and exopodite) parts of developing appendages (*Figure 16C* and *Figure 16—figure supplement 1 E*), which are the limb segments that were missing in the knock-out experiments (*Figure 16B*). Genotyping of one of these embryos demonstrated that the tagged *PhDll-e* locus was indeed encoding a functional *PhDll-e* protein with a small in-frame deletion around the targeted region (*Figure 16—figure supplement 1 F*).

These results, together with the other recent applications of the CRISPR/Cas system to study Hox genes in *Parhyale* (*Martin et al., 2015*; *Serano et al., 2015*), demonstrate that the ability to manipulate the fertilized eggs together with the slow tempo of early cleavages can result in very high targeting frequencies and low levels of mosaicism for both knock-out and knock-in approaches. Considering the usefulness of the genome-wide resources described in this report, we anticipate that the *Parhyale* embryo will prove an extremely powerful system for fast and reliable G0 screens of gene expression and function.

## Conclusion

In this article we described the first complete genome of a malacostracan crustacean species, the genome of the marine amphipod *Parhyale hawaiensis*. At an estimated size of 3.6 Gb, it is among the largest genomes submitted to NCBI. The *Parhyale* genome reported here is that of a single adult male from a sib-bred line called Chicago-F. We find *Parhyale* has an abundance of repetitive sequence and high levels of heterozygosity in the individual sequenced. Combined with analysis of available transcriptome sequences and independently sequenced genomic BAC clones, we conclude high levels of heterozygosity are representative of high levels of single and polynucleotide polymorphisms in the broader laboratory population. Our comparative bioinformatics analyses suggest that the expansion of repetitive sequences and the increase in gene size due to an expansion of intron size have contributed to the large size of the genome. Despite these challenges, the *Parhyale* genome and associated transcriptomic resources reported here provide a useful assembly of most genic regions in the genome and a comprehensive description of the *Parhyale* transcriptome and proteome.

*Parhyale* has emerged since the early 2000's as an attractive animal model for developmental genetic and molecular cell biology research. It fulfills several desirable biological and technical requirements as an experimental model, including a relatively short life-cycle, year-round breeding under standardized laboratory conditions, availability of thousands of eggs for experimentation on a daily basis, and amenability to various embryological, cellular, molecular genetic and genomic approaches. In addition, *Parhyale* has stereotyped cell lineages and cell behaviors, a direct mode of development, a remarkable appendage diversity and the capacity to regenerate limbs post-embryonically. These qualities can be utilized to address fundamental long-standing questions in developmental biology, like cell fate specification, nervous system development, organ morphogenesis and regeneration (*Stamataki and Pavlopoulos, 2016*). Research on these topics will benefit enormously from the standardized genome-wide resources reported here. Forward and reverse genetic analyses using both unbiased screens and candidate gene approaches have already been devised successfully in *Parhyale* (*Table 1*). The availability of coding and non-coding sequences for all identified signaling pathway components, transcription factors and various classes of non-coding RNAs will dramatically

accelerate the study of the expression and function of genes implicated in the aforementioned processes.

Equally importantly, our analyses highlight additional areas where *Parhyale* could serve as a new experimental model to address other questions of broad biomedical interest. From a functional genomics perspective, the *Parhyale* immune system appears to be a good representative of the malacostracan or even the multicrustacean clade that can be studied in detail with existing tools and resources. Besides the evolutionary implications and the characterization of alternative strategies used by arthropods to defend against pathogens, a deeper mechanistic understanding of the *Parhyale* immune system will be relevant to aquaculture. Some of the greatest setbacks in the crustacean farming industry are caused by severe disease outbreaks. *Parhyale* is closely related to farmed crustaceans (primarily shrimps, prawns and crayfish) and the knowledge acquired from studying its innate immunity could help enhance the sustainability of this industry by preventing or controlling infectious diseases (*Stentiford et al., 2012*; *Johnson et al., 2008*; *Lu et al., 2008*; *Rajesh Kumar et al., 2008*; *Rowley and Pope, 2012*).

An immune-related problem that will be also interesting to explore in *Parhyale* concerns the possibility of a sterile digestive tract similar to that proposed for limnoriid Isopods (*King et al., 2010*). *Parhyale*, like limnoriid Isopods, encodes and expresses all enzymes required for lignocellulose digestion, suggesting that it is able to 'digest wood' by itself without symbiotic microbial partners. Of course, a lot of work still needs to be invested in the characterization of the cellulolytic system in *Parhyale* before any comparisons can be made with other well-established symbiotic digestion systems of lignocellulose. Nevertheless, the possibility of an experimentally tractable animal model that serves as a living bioreactor to convert lignocellulose into simpler metabolites, suggests that future research in *Parhyale* may also have a strong biotechnological potential, especially for the production of biofuels from the most abundant and cheapest raw material, plant biomass.

Although more high-quality genomes with a broader phylogenetic coverage are still needed for meaningful evolutionary comparisons, our observations from analysing the *Parhyale* genome and other crustacean data sets also contribute to the ongoing debate on the relationships between crustacean groups. While the analysis of shared orthologous groups did not provide clear support for either the Allotriocarida hypothesis (uniting Branchiopoda with Hexapoda) or the Vericrustacea hypothesis (uniting Branchiopoda with Malacostraca), we noted the presence of GH7 genes and the absence of PGRP genes in branchiopod and multicrustacean genomes supporting the Vericrustacea hypothesis. It still remains to be proven how reliable these two characters will be to distinguish between these alternative phylogenetic affinities.

Finally, *Parhyale* was introduced recently as a new model for limb regeneration (*Konstantinides and Averof, 2014*). In some respects, including the segmented body plan, the presence of a blood system and the contribution of lineage-committed adult stem cells to newly formed tissues, regeneration in *Parhyale* may resemble the process in vertebrates more than other established invertebrate models (e.g. planarians, hydra). Regenerative research in *Parhyale* has been founded on transgenic approaches to label specific populations of cells and will be further assisted by the resources presented here. Likewise, we expect that the new genomic information and CRISPR-based genome editing methodologies together with all other facets of *Parhyale* biology will open other new research avenues not yet imagined.

## Materials and methods

Raw genomic reads are deposited at NCBI with the project accession: PRJNA306836. All supplemental data including IPython notebook can be downloaded from this figshare link: https://figshare.com/articles/supplemental_data_for_Parhyale_hawaniensis_genome/3498104 Alternatively, the IPython notebooks and associated scripts can also be viewed at the following github repository: https://github.com/damiankao/phaw_genome

### Genome library preparation and sequencing

About 10 μg of genomic DNA were isolated from a single adult male from the Chicago-F isofemale line established in 2001 (*Parchem et al., 2010*). The animal was starved for one week and treated for 3 days with penicillin-streptomycin (100x, Gibco/Thermo Fisher Scientific), tetracycline hydrochloride (20 μg/ml, Sigma-Aldrich) and amphotericin B (200x, Gibco/Thermo Fisher Scientific). It was

then flash frozen in liquid nitrogen, homogenized manually with a pestle in a 1.5 ml microtube (Kimble Kontes) in 600 µl of Lysis buffer (100 mM Tris-HCl pH 8, 100 mM NaCl, 50 mM EDTA, 0.5% SDS, 200 µg/ml Proteinase K, 20 µg/ml RNAse A). The lysate was incubated for 3 hr at 37°C, followed by phenol/chloroform extractions and ethanol precipitation. The condensed genomic DNA was fished out with a Pasteur pipette, washed in 70% ethanol, air-dried, resuspended in nuclease-free water and analysed on a Qubit fluorometer (Thermo Fisher Scientific) and on a Bioanalyzer (Agilent Technologies). All genome libraries were prepared from this sample: 1 µg of genomic DNA was used to generate the shotgun libraries using the TruSeq DNA Sample Prep kit (Illumina) combined with size-selection on a LabChip XT fractionation system (Caliper Life Sciences Inc) to yield 2 shotgun libraries with average fragment sizes 431 bp and 432 bp, respectively; 4 µg of genomic DNA were used to generate 4 mate-pair libraries with average fragment sizes 5.5 kb, 7.3 kb, 9.3 kb and 13.8 kb using the Nextera Mate Pair Sample Preparation kit (Illumina) combined with agarose size selection. All libraries were sequenced on a HiSeq 2500 instrument (Illumina) using paired-end 150 nt reads.

## Karyotyping

For chromosome spreads, tissue was obtained from embryos at stages 14–18 (*Browne et al., 2005*). Eggs were taken from the mother and incubated for 1–2 hr in isotonic colchicine solution (0.05% colchicine, artificial sea water). After colchicine incubation, the embryonic tissue was dissected from the egg and placed in hypotonic solution (0.075 M KCl) for 25 min. For tissue fixation, we replaced the hypotonic solution with freshly prepared ice-chilled Carnoy's fixative (six parts ethanol, three parts methanol and one part anhydrous acetic acid) for 25 min. The fixed tissue was minced with a pair of fine tungsten needles in Carnoy's solution and the resulting cell suspension was dropped with a siliconized Pasteur pipette from a height of about 5 cm onto a carefully cleaned ice-chilled microscopic slide. After partial evaporation of the Carnoy's fixative the slides were briefly exposed a few times to hot water vapors to rehydrate the tissue. The slides were then dried on a 75°C metal block in a water bath. Finally, the slides with prepared chromosomes were aged overnight at 60°C. After DNA staining either with Hoechst (H33342, Molecular Probes) or with DAPI (Invitrogen), chromosomes were counted on a Zeiss Axioplan II Imaging equipped with C-Apochromat 63x/1.2 NA objective and a PCO pixelfly camera. FIJI was used to improve image quality (contrast and brightness) and FIJI plugin 'Cell Counter' was used to determine the number of chromosomes.

## Analysis of polymorphism and repetitiveness

The *Parhyale* raw data and assembled data are available on the NCBI website. Genome assembly was done with Abyss (*Simpson et al., 2009*) at two different k-mer settings (70, 120) and merged with GAM-NGS. Scaffolding was performed with SSPACE (*Boetzer et al., 2011*). We chose cut-offs of >95% overlap length and >95% identity when removing shorter allelic contigs before scaffolding as these gave better scaffolding results as assessed by assembly metrics. Transcriptome assembly was performed with Trinity (*Haas et al., 2013*). The completeness of the genome and transcriptome was assessed by blasting against CEGMA genes (*Parra et al., 2007*) and visualized by plotting the orthologue hit ratio versus e-value. K-mer analysis of variant and repetitive branching was performed with String Graph Assembler's preqc module (*Simpson, 2014*). K-mer intersection analysis was performed using jellyfish2 (*Marçais and Kingsford, 2011*). Repetitive elements were annotated with RepeatModeler and RepeatMasker. An in-depth description of the assembly process and repeat masking is detailed in *source code 1* and *2*.

## Transcriptome library preparation, sequencing and assembly

*Parhyale* transcriptome assembly was generated from Illumina reads collected from diverse embryonic stages (Stages 19, 20, 22, 23, 25, and 28), and adult thoracic limbs and regenerating thoracic limbs (3 and 6 days post amputation). For the embryonic samples, RNA was extracted using Trizol; PolyA+ libraries were prepared with the Truseq V1 kit (Illumina), starting with 0.6–3.5 µg of total mRNA, and sequenced on the Illumina Hiseq 2000 as paired-end 100 base reads, at the QB3 Vincent J. Coates Genomics Sequencing Laboratory. For the limb samples, RNA was extracted using Trizol; PolyA+ libraries were prepared with the Truseq V2 kit (Illumina), starting with 1 µg of total mRNA, and sequenced on the Illumina Hiseq 2500 as paired-end 100 base reads, at the IGBMC Microarray and Sequencing platform. 260 million reads from embryos and 180 million reads from limbs were

used for the transcriptome assembly. Prior to the assembly we trimmed adapter and index sequences using cutadapt (*Martin, 2011*). We also removed spliced leader sequences: GAATTTTCACTG TTCCCTTTACCACGTTTTACTG, TTACCAATCACCCCTTTACCAAGCGTTTACTG, CCCTTTACCAAC TCTTAACTG, CCCTTTACCAACTTTACTG using cutadapt with 0.2 error allowance to remove all potential variants (*Douris et al., 2009*). To assemble the transcriptome we used Trinity (version trinityrnaseq_r20140413) (*Haas et al., 2013*) with settings: -min_kmer_cov 2, -path_reinforcement_distance 50.

## Gene model prediction and canonical proteome dataset generation

Gene prediction was done with a combination of Evidence Modeler (*Haas et al., 2008*) and Augustus (*Stanke and Waack, 2003*). The transcriptome was first mapped to the genome using GMAP (*Wu and Watanabe, 2005*). A secondary transcriptome reference assembly was performed with STAR/Cufflinks (*Trapnell et al., 2010*; *Dobin et al., 2013*). The transcriptome mapping and Cufflinks assembly was processed through the PASA pipeline (*Haas et al., 2008*) to consolidate the annotations. The PASA dataset, a set of Exonerate (*Slater and Birney, 2005*) mapped Uniprot proteins, and Ab inito GeneMark (*Lukashin and Borodovsky, 1998*) predictions were consolidated with Evidence Modeler to produce a set of gene annotations. A high confidence set of gene models from Evidence Modeler containing evidence from all three sources was used to train Augustus. Evidence from RepeatMasker (*Smit et al., 2013*, PASA and Exonerate were then used to generate Augustus gene predictions. A final list of genes for down-stream analysis was generated using both transcriptome and gene predictions (canonical proteome dataset). Detailed methods are described in *Source code 3*.

## Polymorphism analysis on genic regions and BAC clones

For variant analysis on the BAC clones, the short shot-gun library genomic reads were mapped to the BAC clones individually. GATK was then used to call variants. For variant analysis on the genic regions, transcript sequences used to generate the canonical proteome dataset were first aligned to the genome assembly. Genome alignments of less than 30 base pairs were discarded. The possible genome alignments were sorted based on number of mismatches with the top alignment having the least amount of mismatches. For each transcript, the top two genome aligments were used to call potential variants. Trascripts or parts of transcripts where there were more than five genomic mapping loci were discarded as potentially highly conserved domains or repetitive regions. Detailed methods of this process are described in *Source code 4*.

## Polymorphisms in *Parhyale* developmental genes

*Parhyale* genes (nucleotide sequences) were downloaded from GenBank. Each gene was used as a query for blastn against the *Parhyale* genome using the Geneious software (*Kearse et al., 2012*). In each case two reference contig hits were observed where both had E values of close to zero. A new sequence called geneX_snp was created and this sequence was annotated with the snps and/or indels present in the alternative genomic contigs. To determine the occurrence of synonymous and non-synonymous substitutions, the original query and the newly created sequence (with polymorphisms annotated) were in silico translated into protein sequences followed by pairwise alignment. Regions showing amino acid changes were annotated as non-synonymous substitutions. Five random genes from the catalogue were selected for PCR, cloning and Sanger sequencing to confirm genomic polymorphisms and assess further polymorphism in the lab popultaion. Primers for genomic PCR designed to capture and amplify exon regions are listed as the following: dachshund (PH1F = 5'- GG TGCGCTAAATTGAAGAAATTACG-3' and PH1R = 5'- ACTCAGAGGGTAATAGTAACAGAA-3'), distalless exon 2 (PH2F = 5'-CACGGCCCGGCACTAACTATCTC-3' and PH2R = 5'-GTAATATATCTTA-CAACAACGACTGAC-3'), distalless exon 3 (PH3F = 5'-GGTGAACGGGCCGGAGTCTC-3' and PH3R = 5'-GCTGTGGGTGCTGTGGGT-3'), homothorax (PH4F = 5'-TCGGGGTGTAAAAAGGACTCTG-3' and PH4R = 5'-AACATAGGAACTCACCTGGTGC-3'), orthodenticle (PH5F = 5'-TTTGCCACTAA-CACATATTTCGAAA-3' and PH5R = 5'-TCCCAAGTAGATGATCCCTGGAT-3') and prospero (PH6F = 5'-TACACTGCAACATCCGATGACTTA-3' and PH6R = 5'-CGTGTTATGTTCTCTCGTGGCTTC-3').

## Evolutionary analyses of orthologous groups

Evolutionary analyses and comparative genomics were performed with 16 species: *D. melanogaster, A. gambiae, D. pulex, L. salmonis, S.maritima, S. mimosarum, M. martensii, I. scapularis, H. dujardini, C. elegans, B. malayi, T. spiralis, M. musculus, H. sapiens*, and *B. floridae*. For orthologous group analyses, gene families were identified using OrthoFinder (*Emms and Kelly, 2015*). The canonical proteome was used as a query in BlastP against proteomes from 16 species to generate a distance matrix for OrthoFinder to normalize and then cluster with MCL. Detailed methods are described in *Source code 5*. For the comparative BLAST analysis, five additional transcriptome datasets were used from the following crustacean species: *Litopenaeus vannamei, Echinogammarus veneris, Eucyclops serrulatus, Calanus finmarchicus, Speleonectes tulumensis.*

## Fluorescence in situ hybridization detection of Hox genes

Embryo fixation and in-situ hybridization was performed according to (*Rehm et al., 2009*). To enhance the nascent nuclear signal over mature cytoplasmic transcript, we used either early germ-band embryos (Stages 11 – 15) in which expression of *lab, Dfd*, and *Scr* are just starting (*Serano et al., 2015*), or probes that contain almost exclusively intron sequence (*Ubx, abd-A, Abd-B*, and *en1*). *Lab, Dfd*, and *Scr* probes are described in (*Serano et al., 2015*). Template for the intron-spanning probes were amplified using the following primers: en1-Intron1, AAGACACGAC-GAGCATCCTG and CTGTGTATGGCTACCCGTCC; Ubx-Intron1, GGTATGACAGCCGTCCAACA and AGAGTGCCAAGGATACCCGA; abd-A, CGATATACCCAGTCCGGTGC and TCATCAGC-GAGGGCACAATT; Abd-B, GCTGCAGGATATCCACACGA and TGCAGTTGCCGCCATAGTAA.

A T7-adapter was appended to the 5' end of each reverse primer to enable direct transcription from PCR product. Probes were labeled with either Digoxigenin (DIG) or Dinitrophenol (DNP) conjugated UTPs, and visualized using sheep -DIG (Roche) and donkey -Sheep AlexaFluor 555 (Thermo Fischer Scientific), or Rabbit -DNP (Thermo Fischer Scientific) and Donkey -Rabbit AlexaFluor 488 (Jackson ImmunoResearch), respectively. Preparations were imaged on an LSM 780 scanning laser confocal (Zeiss), and processed using Volocity software (Perkin-Elmer).

## Cross species identification of GH family genes and immune-related genes

The identification of GH family genes was done by obtaining Pfam annotations (*Finn et al., 2006*) for the *Parhyale* canonical proteome. Pfam domains were classified into different GH families based on the CAZy database (*Cantarel et al., 2009*). For immune-related genes, best-reciprocal blast was performed with ImmunoDB genes (*Waterhouse et al., 2007*).

## Phylogenetic tree construction

Multiple sequence alignments of protein sequences for gene families of *FGF, FGFR, CERS, GH7, GH9, PGRP*, Toll-like receptors, *DICER*, Piwi and Argonaute were performed using MUSCLE (*Edgar, 2004*). Phylogenetic tree construction was performed with RAxML (*Stamatakis, 2014*) using the WAG+G model from MUSCLE multiple alignments.

## Bisulfite sequencing

Libraries for DNA methylation analysis by bisulfite sequencing were constructed from 100ng of genomic DNA extracted from one *Parhyale* male individual, using the Illumina Truseq DNA methylation kit according to manufacturers instructions. Alignments to the *Parhyale* genome were generated using the core Bismark module from the program Bismark (*Krueger and Andrews, 2011*), having first artificially joined the *Parhyale* contigs to generate 10 pseudo-contigs as the program is limited as to the number of separate contigs it can analyse. We then generated genome-wide cytosine coverage maps using the bismark_methylation_extraction module with the parameter 'CX specified to generate annotations of CG, CHH and CHG sites. In order to analyse genome-wide methylation patterns only cytosines with more than a 10 read depth of coverage were selected. Overall methylation levels at CG, CHH and CHG sites were generated using a custom Perl script. To analyse which regions were methylated we mapped back from the joined contigs to the original contigs and assigned these to functional regions based on RepeatMasker (*Smit et al., 2013*) and transcript annotations of repeats and genes respectively. To generate overall plots of methylation levels in different

features we averaged over all sites mapping to particular features, focusing on CG methylation and measuring the% methylation at each site as the number of reads showing methylation divided by the total number of reads covering the site. Meta gene plots over particular features were generated similarly except that sites mapping within a series of 100 bp wide bins from 1000 bp upstream of the feature start site and onward were collated.

## Identification and cloning of Dscam alternative spliced variants

For the identification of *Dscam* in the *Parhyale*, we used the Dscam protein sequence from crustaceans *D. pulex* (*Brites et al., 2008* ) and *L. vannamei* (*Chou et al., 2009-12*) as queries to probe the assembled genome using tBlastN. A 300kb region on scaffold phaw_30.0003392 was found corresponding to the *Parhyale Dscam* extending from IG1 to FN6 exons. This sequence was annotated using transcriptome data together with manual searches for open reading frames to identify IG, FN exons and exon-intron boundaries (*Figure 13—supplemental figure 1*). Hypervariable regions of IG2, IG3 and IG7 were also annotated accordingly on the scaffold (*Figure 13—supplemental figure 1*). This region represents a bona fide *Dscam* paralog as it matches the canonical extracellular *Dscam* domain structure of nine IGs – four FNs – one IG and two FNs. *Parhyale* mRNA extractions were performed using the Zymo Research Direct-zol RNA MiniPrep kit according to manufacturer's instructions. Total RNA extract was used for cDNA synthesis using the Qiagen QuantiTect Reverse Transcription Kit according to manufacturer's instructions. To identify and confirm potential hypervariable regions from the *Parhyale* (Ph-Dscam) transcript, three regions of Ph-Dscam corresponding to IG2, IG3 and IG7 exons respectively were amplified using the following primer pairs. IG2 region:

DF1 = 5'-CCCTCGTGTTCCCGCCCTTCAAC-3'
DR1 = 5'-GCGATGTGCAGCTCTCCAGAGGG-3'
IG3 region:
DF2 = 5'-TCTGGAGAGCTGCACATCGCTAAT-3'
DR2 = 5'-GTGGTCATTGCGTACGAAGCACTG-3'
IG7 region:
DF3 = 5'-CGGATACCCCATCGACTCCATCG-3'
DR3 = 5'-GAAGCCGTCAGCCTTGCATTCAA-3'

PCR of each region was performed using Phusion High-fidelity polymerase from Thermo Fisher Scientific and thermal cycling was done as the following: 98C 30s, followed by 30 cycles of 98C 10s, 67C 30s, 72C 1m30s, and then 72C 5m. PCR products were cloned into pGEMT-Easy vector and a total of 81 clones were selected and Sanger sequenced and in silico translated in the correct reading frame using Geneious (R7; (*Kearse et al., 2012*) for multiple sequence alignment.

## Identification of non-protein-coding RNAs

*Parhyale* non-protein-coding RNAs were identified using two independent approaches. Infernal 1.1.1 (*Nawrocki and Eddy, 2013*) was used with the RFAM 12.0 database (*Nawrocki et al., 2015*) to scan the genome to identify potential non-coding RNAs. Additionally, MiRPara (*Wu et al., 2011*) was used to scan the genome for potential miRNA precursors. These potential precursors were further filtered using small RNA read mapping and miRBase mapping (*Griffiths-Jones et al., 2008*). Putative lncRNAs were identified from the transcriptome by applying filtering criteria including removal of known and predicted coding RNAs. Detailed methods are available in Supplementary Data 11.

## CRISPR/Cas genome editing

To genotype our wild-type population, extraction of total RNA and preparation of cDNA from embryos were carried out as previously described (*Pavlopoulos et al., 2009*). The PhDll-e cDNA was amplified with primers PhDlle_2For (5'-TTTGTCAGGGATCTGCCATT-3') and PhDlle_1852Rev (5'-TAGCGGCTGACGGTTGTTAC-3'), purified with the DNA Clean and Concentrator kit (Zymo Research), cloned with the Zero Blunt TOPO PCR Cloning Kit (Thermo Fisher Scientific) and sequenced with primers M13 forward (5'- GTAAAACGACGGCCAG-3') and M13 reverse (5'- CAGGAAACAGCTATGAC-3').

Each template for sgRNA synthesis was prepared by annealing and PCR amplification of the sgRNA-specific forward primer Dll1: (18 nt PhDll-e-targeted sequence underlined)

5'-GAAATTAATACGACTCACTATA

AGAGTTGTTACCAAAGAAGTTTTAGAGCTAGAAATAGC-3'
or Dll2: (20 nt PhDll-e-targeted sequence underlined)
5'-GAAATTAATACGACTCACTAT
AGGCTTCCCCGCCGCCATGTAGTTTTAGAGCTAGAAATAGC-3'
together with the universal reverse primer:
5'-AAAAGCACCGACTCGGTGCCACTTTTTCAAGTTGATAA
CGGACTAGCCTTATTTTAACTTGCTATTTCTAGCTCTAAAAC-3'
using the Phusion DNA polymerase (New England Biolabs).

Each PCR product was gel-purified with the Zymoclean DNA recovery kit (Zymo Research) and 150 ng of DNA were used as template in an in vitro transcription reaction with the Megashortscript T7 kit (Thermo Fisher Scientific). A 4-hr incubation at 37°C was followed by DNAse digestion, phenol/chloroform extraction, ethanol precipitation and storage in ethanol at −20° C according to the manufacturer's instructions. Before microinjection, a small aliquot of the sgRNA was centrifuged, the pellet was washed with 70% ethanol, resuspended in nuclease-free water and quantified on a Nanodrop spectrophotometer (Thermo Scientific). The Cas9 was provided either as in vitro synthesized caped mRNA or as recombinant protein. Cas9 mRNA synthesis was carried out as previously described (*Kontarakis and Pavlopoulos, 2014*) using plasmid T7-Cas9 (a gift from David Stern and Justin Crocker) linearized with EcoRI digestion. The lyophilized Cas9 protein (PNA Bio Inc) was resuspended in nuclease-free water at a concentration of 1.25 µg/µl and small aliquots were stored at −80°C. For microinjections, we mixed 400 ng/µl of Cas9 protein with 40–200 ng/µl sgRNA, incubated at 37°C for 5 min, transferred on ice, added the inert dye phenol red (5x from Sigma-Aldrich) and, for knock-in experiments, the tagging plasmid at a concentration of 10 ng/µl. The injection mix was centrifuged for 20 min at 4°C and the cleared solution was microinjected into 1-cell-stage embryos as previously described (*Kontarakis and Pavlopoulos, 2014*).

In the knock-out experiments, embryos were scored for phenotypes under a bright-field stereomicroscope 7–8 days after injection (stage S25-S27) when organogenesis is almost complete and the limbs are clearly visible through the transparent egg shell. To image the cuticle, anaesthetized hatchlings were fixed in 2% paraformaldehyde in 1xPBS for 24 hr at room temperature. The samples were then washed in PTx (1xPBS containing 1% TritonX-100) and stained with 1 mg/ml Congo Red (Sigma-Aldrich) in PTx at room temperature with agitation for 24 hr. Stained samples were washed in PTx and mounted in 70% glycerol for imaging. Serial optical sections were obtained at 2 µm intervals with the 562 nm laser line on a Zeiss 710 confocal microscope using the Plan-Apochromat 10x/0.45 NA objective. Images were processed with Fiji (http://fiji.sc) and Photoshop (Adobe Systems Inc).

This methodology enabled us to also extract genomic DNA for genotyping from the same imaged specimen. Each specimen was disrupted with a disposable pestle in a 1.5 ml microtube (Kimble Kontes) in 50 µl of Squishing buffer (10 mM Tris-HCl pH 8, 1 mM EDTA, 25 mM NaCl, 200 µg/ml Proteinase K). The lysate was incubated at 37°C for a minimum of 2 hr, followed by heat inactivation of the Proteinase K for 5 min at 95°C, centrifugation at full speed for 5 min and transferring of the cleared lysate to a new tube. To recover the sequences in the PhDll-e locus targeted by the Dll1 and Dll2 sgRNAs, 5 µl of the lysate were used as template in a 50 µl PCR reaction with the Phusion DNA polymerase (New England Biolabs) and primers 313For (5'-TGGTTTTAGCAACAGTGAAGTGA-3') and 557Rev (5'-GACTGGGAGCGTGAGGGTA-3'). The amplified products were purified with the DNA Clean and Concentrator kit (Zymo Research), cloned with the Zero Blunt TOPO PCR Cloning Kit (Thermo Fisher Scientific) and sequenced with the M13 forward primer.

For the knock-in experiments, we constructed the tagging plasmid pCRISPR-NHEJ-KI-Dll-T2A-H2B-Ruby2 that contained the PhDll-e coding sequence fused in-frame with the T2A self-cleaving peptide, the *Parhyale histone* H2B and the Ruby 2 monomeric red fluorescent protein, followed by the PhDll-e 3'UTR and the pGEM-T Easy vector backbone (Promega). This tagging plasmid has a modular design with unique restriction sites for easy exchange of any desired part. More details are available upon request. Embryos co-injected with the Cas9 protein, the Dll2 sgRNA and the pCRISPR-NHEJ-KI-Dll-T2A-H2B-Ruby2 tagging plasmid were screened for nuclear fluorescence in the developing appendages under an Olympus MVX10 epi-fluorescence stereomicroscope. To image expression, live embryos at stage S22 were mounted in 0.5% SeaPlaque low-melting agarose (Lonza) in glass bottom microwell dishes (MatTek Corporation) and scanned as described above acquiring both the fluorescence and transmitted light on an inverted Zeiss 880 confocal microscope.

To recover the chromosome-plasmid junctions, genomic DNA was extracted from transgenic siblings with fluorescent limbs and used as template in PCR reaction as described above with primer pair 313For and H2BRev (5'-TTACTTAGAAGAAGTGTACTTTG-3') for the left junction and primer pair M13 forward and 557Rev for the right junction. Amplified products were purified and cloned as described above and sequenced with the M13 forward and M13 reverse primers.

## Acknowledgements

We are grateful to Serge Picard for sequencing the genome libraries, and Frantisek Marec and Peer Martin for useful advice on *Parhyale* karyotyping.

## Additional information

### Funding

| Funder | Grant reference number | Author |
|---|---|---|
| Biotechnology and Biological Sciences Research Council | BB/K007564/1 | Damian Kao<br>Alvina G Lai<br>Alessia Di Donfrancesco<br>Natalia Pouchkina-Stancheva<br>Aziz Aboobaker |
| Medical Research Council | MR/M000133/1 | Damian Kao<br>Alvina G Lai<br>Alessia Di Donfrancesco<br>Natalia Pouchkina-Stancheva<br>Aziz Aboobaker |
| Human Frontier Science Program | | Alvina G Lai |
| University of Oxford | Elizabeth Hannah Jenkinson Fund | Alvina G Lai<br>Aziz Aboobaker |
| Howard Hughes Medical Institute | | Evangelia Stamataki<br>Suyash Kumar<br>Igor Siwanowicz<br>Andy Le<br>Andrew Lemire<br>Anastasios Pavlopoulos |
| Medical Research Council | MRC MC-A652- 5PZ80 | Silvana Rosic<br>Peter Sarkies |
| Agence Nationale de la Recherche | ANR-12-CHEX-0001-01 | Nikolaos Konstantinides<br>Marco Grillo<br>Michalis Averof |
| National Science Foundation | IOS-1257379 | Erin Jarvis<br>Heather Bruce<br>Nipam H Patel |
| Imperial College London | | Peter Sarkies |
| John Fell Fund, University of Oxford | | Aziz Aboobaker |

The funders had no role in study design, data collection and interpretation, or the decision to submit the work for publication.

### Author contributions

DK, Devised assembly strategy, Assembled and analyzed the sequencing data, Annotated the genome, transcriptome and proteome, Performed orthology group analysis, Annotated small RNAs, Drafting and revising the article; AGL, Analysed the genome including major signaling pathways, polymorphisms, immunity, lignocellulose digestion, epigenetic pathways, small RNA pathways and small RNAs, Cloning of DSCAM variants, Experimental confirmation of polymorphisms, Drafting and revising the article; ES, Prepared the genomic libraries, performed CRISPR knock-out, performed CRISPR knock-in, Drafting and revising the article; SR, Contributed bisulfite sequencing data and

analysis of genome wide methylation; NK, MS, MG, MA, Contributed transcriptome data and transcriptome assembly; EJ, Performed in situ hybridization detection of Hox genes, and interpreted data; ADD, NP-S, Contributed to confirmation of polymorphism and cloning of Ph-DSCAM variants; HB, Contributed transcriptome data; SK, Performed CRISPR knock-out; IS, Performed Parhyale cuticle staining; ALe, Performed CRISPR knock-in; ALemi, Was consulted about sequencing strategy and helped with bioinformatics; MBE, Contributed to RNAseq data production; CE, Contributed to project planning; WEB, Established the Chicago-F line; CW, Performed karyotyping; NHP, Performed in situ hybridization detection of Hox genes, and interpreted data. Contributed transcriptome data, Established the Chicago-F line; PS, Conceived and designed bisulfite sequencing experiments, contributed bisultfite sequencing data and analysis of genome wide methylation; AP, Conceived, designed and managed the project, Contributed to data acquisition and analysis, Drafting and revising the article; AA, Devised assembly strategy, Contributed to data analysis, Conceived, designed and managed the project, Drafting and revising the article

### Author ORCIDs
Alvina G Lai, http://orcid.org/0000-0001-8960-8095
Suyash Kumar, http://orcid.org/0000-0002-5861-4027
Cassandra Extavour, http://orcid.org/0000-0003-2922-5855
Carsten Wolff, http://orcid.org/0000-0002-5926-7338
Michalis Averof, http://orcid.org/0000-0002-6803-7251
Aziz Aboobaker, http://orcid.org/0000-0003-4902-5797

## Additional files

### Supplementary files

• Source code 1. iPython Notebook for *Parhyale* genome assembly. Includes bioinformatic processsing of raw read data, k-mer analysis, contig assembly, scaffolding and CEGMA cased representation analyis.

• Source code 2. iPython Notebook for repeat analysis. Includes repeat analysis of the Parhyale genome using Repeat Modeller and Repeat Masker.

• Source code 3. iPython Notebook for transcriptome and annotation. *Parhyale* transcriptome assembly, genome annotation and generation of canonical proteome dataset.

• Source code 4. iPython Notebook for variant analysis. Analysis of polymorphism in *Parhyale* using genome reads, transcriptome data and sanger sequenced BACs.

• Source code 5. iPython Notebook of orthology analysis. Protein orthology analysis between *Parhyale* and other species

• Source code 6. iPython Notebook for RNA. Analysis of microRNAs and putative lncRNAs in *Parhyale*.

### Major datasets

The following datasets were generated:

| Author(s) | Year | Dataset title | Dataset URL | Database, license, and accessibility information |
|---|---|---|---|---|
| Damian Kao, Alvina G Lai, Aziz Aboobaker | 2016 | Parhyale hawaiensis isolate: isofemale 4 Genome sequencing | http://www.ncbi.nlm.nih.gov/bioproject/306836 | Publicly available at the NCBI BioProject database (accession no: PRJNA306836) |

| Silvana Rosic, Peter Sarkies, Aziz Aboobaker | 2016 | Bisulfite sequencing of Parhyale hawaiensis | https://www.ncbi.nlm.nih.gov/geo/query/acc.cgi?acc=GSE82141 | Publicly available at the NCBI Gene Expression Omnibus (accession no: GSE82141) |

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
