## [Decision Letter]

Thank you for submitting your article "The genome of *Parhyale hawaiensis*: a model for animal development, regeneration, immunity and ligno-cellulose digestion" for consideration by *eLife*. Your article has been reviewed by three peer reviewers, including Heinrich Reichert (Reviewer #2), and the evaluation has been overseen by K VijayRaghavan as the Reviewing Editor and Senior Editor.

The reviewers have discussed the reviews with one another and the Reviewing Editor has drafted this decision to help you prepare a revised submission.

General comments:

For long, our understanding of biology has advanced through studies in 'model' organisms that rode wave of the molecular revolution. Recent advances in imaging and genome editing hold a similar promise for other organisms that offer great potential for fundamental discoveries in cellular and developmental biology and an understanding of biology more representative of the diversity of life on Earth.

Within the last two decades, the crustacean *Parhyale hawaiensis*, has emerged as one such new model organism, particularly for studying limb development and postembryonic limb regeneration. This is due to favorable technical and biological features such as short life cycle, year round breeding in the lab and the production of large numbers of eggs for in vivo experimentation. However, in contrast to the other leading crustacean model, the water flea *Daphnia*, a complete genome sequence and corresponding annotation are lacking in *Parhyale*, and this lack is currently a major stumbling block for rapid experimental progress in this malacostracan system. In this light, its genome, described in this paper, has been an eagerly awaited resource.

In this paper, the authors have sequenced, analysed and annotated the genome of *Parhyale hawaiensis*, which belongs to the malacostracan clade of pancrustaceans. Significantly, it is among the first few non-hexapod genomes to be sequenced, which, given the vast diversity of the arthropods needs to be explored further. Important features of this genome include:

• Its large size. At 3.6GB, is also among the largest genomes to be sequenced.

• Presence of repetitive sequences (including transposable elements), which contributes to the large genome size.

• High levels of heterozygosity.

• Presence of enzymes for lignocellulose digestion.

• Presence of a wide repertoire of genes related to innate immunity.

• Presence of non-coding RNAs and pathways that process them.

• Significant levels of methylation, especially at the sites of transposable elements suggesting epigenetic silencing of regions of the genome.

Importantly, the *Parhyale hawaiensis* genome will be an invaluable resource for functional genomic studies in a crustacean species that has otherwise already been established as a model laboratory organism. The authors demonstrate the power of this by using the CRISPR/Cas9 technology to knock-out the function of, as well as knock-in a tag into the distalless gene, which is involved in limb development.

Specific comments:

The high level of heterozygosity in the genome is particularly striking. Does the use of an isogenised line affect the assessment of this heterozygosity? Does the high heterozygosity in turn affect the methods of sequencing/assembly, and if so, do special techniques have to be employed for these processes? For example, the authors show that regions of high coverage show low heterozygosity whereas regions of low coverage show high levels of heterozygosity. However, data from transcriptome analysis finds no such differences heterozygosity between these groups. While the authors see this as an attribute of read coverage in divergent regions, could the inverse be true? Could the assessment of heterozygosity be exaggerated due to lower coverage? We are sure the authors have considered these possibilities, and it might be helpful if they were discussed in the paper for the broad audience of *eLife* readership.

While the Introduction seems to suggest that this genome will help solve the disputed phylogeny of the pan crustaceans, in fact the phylogeny is still ambiguous, we presume because high quality genome data does not exist for other members. (In the Discussion, however, the authors favour one hypothesis.) It might be valuable if the authors instead stated explicitly that this data is not sufficient to sort out the phylogeny and more genomes need to be sequenced to work it out.

This paper is written from a functional genomics perspective and in general we think the authors shy away from making definitive statements/discussing their findings from an evolutionary perspective. For example, the authors do perform a comparative analysis with proteomes. However, when they discuss individual protein families, they don't comment on the evolutionary ramifications, except for a few cases (GH7 and PGRPs). While we understand that often no evolutionary conclusions might be possible, leaving the matter unaddressed makes the writing feel like it's very much on the backfoot and leaves the reader expecting a discussion that never came!

---

## [Author Response]

*[…] Specific comments:*

*The high level of heterozygosity in the genome is particularly striking. Does the use of an isogenised line affect the assessment of this heterozygosity?*

In this study, we sequenced a single adult male specimen from a line (called Chicago-F) derived from a single female after two rounds of sib mating. Specifically, F1 offspring from a single female were interbred, and from this collection of siblings a single gravid female was removed and its F2 offspring in turn interbred. A single F2 gravid female was then selected and from this single brood the Chicago-F line was established.

We have added the following text to clarify this:

“Sequencing was performed on genomic DNA isolated from a single adult male taken from a line derived from a single female and expanded after two rounds sib-mating”.

This level of inbreeding will not greatly reduce levels of starting heterozygosity, but equally the sequence of one individual alone tell us a little about population level polymorphism. Our primary goal was the de novo assembly of the very complex *Parhyale* genome, rather than the assessment of population genetic parameters. Nevertheless, in order to get a better view of polymorphism levels in our laboratory populations, we also analysed genomic data in comparison to previously sequenced BAC clones from the same Chicago-F strain, and transcriptomic data and targeted Sanger sequencing from the broader lab population. These analyses are described in the section “High levels of heterozygosity and polymorphism in the *Parhyale* genome”.

*Does the high heterozygosity in turn affect the methods of sequencing/assembly, and if so, do special techniques have to be employed for these processes?*

In order to clarify our approach we have added text and the following citation:

“Different strategies have been employed to sequence highly heterozygous diploid genomes of non-model and wild-type samples (Kajitani R. et al., Efficient de novo assembly of highly heterozygous genomes from whole-genome shotgun short reads, Genome Research 24:1384–1395).”

We have also slightly rewritten the description of our assembly process to be more accessible in the current version:

“We aimed for an assembly representative of different haplotypes, allowing manipulations to be targeted to different allelic variants in the assembly. […] We then kept the longer contig of each pair for scaffolding using our mate-pair libraries (Figure 3), after which we added back the shorter allelic contigs to produce the final genome assembly (Figure 4).”

*For example, the authors show that regions of high coverage show low heterozygosity whereas regions of low coverage show high levels of heterozygosity. However, data from transcriptome analysis finds no such differences heterozygosity between these groups. While the authors see this as an attribute of read coverage in divergent regions, could the inverse be true? Could the assessment of heterozygosity be exaggerated due to lower coverage? We are sure the authors have considered these possibilities, and it might be helpful if they were discussed in the paper for the broad audience of eLife readership.*

To clarify, regions of the genome with higher heterozygosity between alleles assemble as separate contigs and regions with lower heterozygosity between alleles as a single contig. The actual distribution of heterozygosity is a continuous variable but the nature of both the assembly and mapping algorithms separates regions of higher and lower heterozygosity. Genome wide heterozygosity is considered in two places in our work, during k-mer analysis before assembly and during our scaffolding process. To make things clearer, in addition to the edits above, we have edited the text as follows:

“The k-mer analysis revealed a bimodal distribution of error-free k-mers (Figure 3). The higher-frequency peak corresponded to k-mers present on both haplotypes (i.e. homozygous regions), while the lower-frequency peak had half the coverage and corresponded to k-mers present on one haplotype (i.e. heterozygous regions) (Simpson and Durbin, 2011). We concluded that the single sequenced adult *Parhyale* exhibits very high levels of heterozygosity, similar to the highly heterozygous oyster genome (see below)."

We then turn our attention from the whole assembly to gene loci (transcribed regions) identified on the assembled contigs. In remapping genomic reads to these regions we found that, not surprisingly, genes with lower heterozygosity had higher number of mapped reads (150x). This is because reads from both haplotypes in the sequenced individual can map to these genes. Conversely for genes with higher heterozygosity only reads from the same haplotype map, and hence they have approximately half the coverage (75x). This of course is very similar to the assembly process and k-mer analysis presented earlier. To clarify our method and explain our findings we have rewritten the text as follows:

“To estimate the level of heterozygosity in genes we first identified transcribed regions of the genome by mapping back transcripts in the assembly. […] Thus, we conclude that heterozygosity influences read mapping and assembly of transcribed regions, and not just non-coding parts of the assembly”

Similarly we have rewritten the text concerning analysis of polymorphism in transcribed regions amongst the wider laboratory population so it explains our findings with greater clarity:

“The assembled *Parhyale* transcriptome was derived from various laboratory populations, hence we expected to see additional polymorphisms beyond those detected in the two haplotypes of the individual male we sequenced. […] For example, we found that the cDNAs of the germ line determinants, nanos (78 SNPS, 34 non-synonymous substitutions and one 6bp indel) and vasa (37 SNPs, 7 non-synonymous substitutions and one 6bp indel) can have more variability within laboratory *Parhyale* populations than might be observed for orthologs between closely related species.”

*While the Introduction seems to suggest that this genome will help solve the disputed phylogeny of the pan crustaceans, in fact the phylogeny is still ambiguous, we presume because high quality genome data does not exist for other members. (In the Discussion, however, the authors favour one hypothesis.) It might be valuable if the authors instead stated explicitly that this data is not sufficient to sort out the phylogeny and more genomes need to be sequenced to work it out.*

*This paper is written from a functional genomics perspective and in general we think the authors shy away from making definitive statements/discussing their findings from an evolutionary perspective. For example, the authors do perform a comparative analysis with proteomes. However, when they discuss individual protein families, they don't comment on the evolutionary ramifications, except for a few cases (GH7 and PGRPs). While we understand that often no evolutionary conclusions might be possible, leaving the matter unaddressed makes the writing feel like it's very much on the backfoot and leaves the reader expecting a discussion that never came!*

We agree that the paucity of high-quality assembled genomes from the non-hexapod pancrustacean lineages does not allow any final conclusions to be drawn regarding their phylogenetic relationships. Although, we present certain similarities between Branchiopoda and Malacostraca in terms of gene gain/loss, currently we do not favour either the Allotriocarida or Vericrustacea hypothesis. Therefore, we have modified the text in the Introduction as follows:

“The genome of the amphipod crustacean *Parhyale hawaiensis* addresses the paucity of high quality non-hexapod genomes among the pancrustacean group, and will help to resolve relationships within this group as more genomes and complete proteomes become available (Rivarola-Duarte et al., 2014, Kenny et al., 2014).”

From the immunity Results section we have deleted the lines:

“Interestingly, the loss of PGRPs and presence of GH7 genes in Branchiopoda, similar to the presence of GH7 genes, supports their close relationship with the Multicrustacea rather than the Hexapoda.”

In the Discussion we have modified text as follows:

“Although more high-quality genomes with a broader phylogenetic coverage are still needed for meaningful evolutionary comparisons, our observations from analysing the *Parhyale* genome and other crustacean data sets also contribute to the ongoing debate on the relationships between crustacean groups. […] It still remains to be proven how reliable these two characters will be to distinguish between these alternative phylogenetic affinities.”

We hope this is a more accurate reflection of our expectations and findings and improves the accessibility of the manuscript.